# Tandem microplastic degradation and hydrogen production by hierarchical carbon nitride-supported single-atom iron catalysts

Jingkai Lin[1,5], Kunsheng Hu[1,5], Yantao Wang[1], Wenjie Tian [1,2] ✉, Tony Hall [3], Xiaoguang Duan [1], Hongqi Sun [4], Huayang Zhang [1,2] ✉, Emiliano Cortés [2] & Shaobin Wang [1] ✉

Microplastic pollution, an emerging environmental issue, poses significant threats to aquatic ecosystems and human health. In tackling microplastic pollution and advancing green hydrogen production, this study reveals a tandem catalytic microplastic degradation-hydrogen evolution reaction (MPD-HER) process using hierarchical porous carbon nitride-supported single-atom iron catalysts (FeSA-hCN). Through hydrothermal-assisted Fenton-like reactions, we accomplish near-total ultrahigh-molecular-weight-polyethylene degradation into $C_3$-$C_{20}$ organics with 64% selectivity of carboxylic acid under neutral pH, a leap beyond current capabilities in efficiency, selectivity, eco-friendliness, and stability over six cycles. The system demonstrates versatility by degrading various daily-use plastics across different aquatic settings. The mixture of FeSA-hCN and plastic degradation products further achieves a hydrogen evolution of 42 μmol $h^{-1}$ under illumination, outperforming most existing plastic photoreforming methods. This tandem MPD-HER process not only provides a scalable and economically feasible strategy to combat plastic pollution but also contributes to the hydrogen economy, with far-reaching implications for global sustainability initiatives.

Plastic pollution is a pervasive and growing problem[1]. In 2022, worldwide plastic production soared to 400.1 million tons, with over 80% ending up being improperly disposed of, contaminating ecosystems through land, waterways, and the atmosphere (see Fig. 1, left)[2]. The COVID-19 pandemic further exacerbated this issue, generating an additional 8 million tons of plastic waste[3–5]. As a particularly troubling aspect of plastic pollution, microplastics (MPs, particles less than 5 mm in size) have infiltrated the environment, creating persistent ecological challenges (center of Fig. 1), due to their long residence time, fast diffusion, and ease of ingestion by organisms[6,7]. It is practically challenging to manually remove or recycle MPs from the environment due

to their small sizes and limited visibility[8]. The intricate compositions that include cross-linkers, antioxidants, or fillers, make the recycling or upcycling of existing (micro)plastics more complex[9].

For (micro)plastics upcycling, recent works have explored photocatalytic[10,11], electrocatalytic[12] and photoelectrochemical[13] methods for manipulating MPs while also generating hydrogen ($H_2$). However, these techniques involve pre-treatment by strong acid or base for plastic hydrolysis, which can pose environmental hazards, safety risks, and high operational costs due to corrosive chemical handling, waste generation, and equipment corrosion. Moreover, only a few plastics, i.e., poly(lactic acid) and poly(ethylene terephthalate),

[1]School of Chemical Engineering, The University of Adelaide, North Terrace, Adelaide, SA 5005, Australia. [2]Nano-Institute Munich, Faculty of Physics, Ludwig-Maximilians-Universität München, Munich, Germany. [3]Mawson Analytical Spectrometry Services, Faculty of Sciences, Engineering and Technology, The University of Adelaide, Adelaide, SA 5005, Australia. [4]School of Molecular Sciences, The University of Western Australia, Perth, WA 6009, Australia. [5]These authors contributed equally: Jingkai Lin, Kunsheng Hu. ✉e-mail: wenjie.tian@adelaide.edu.au; huayang.zhang@adelaide.edu.au; shaobin.wang@adelaide.edu.au

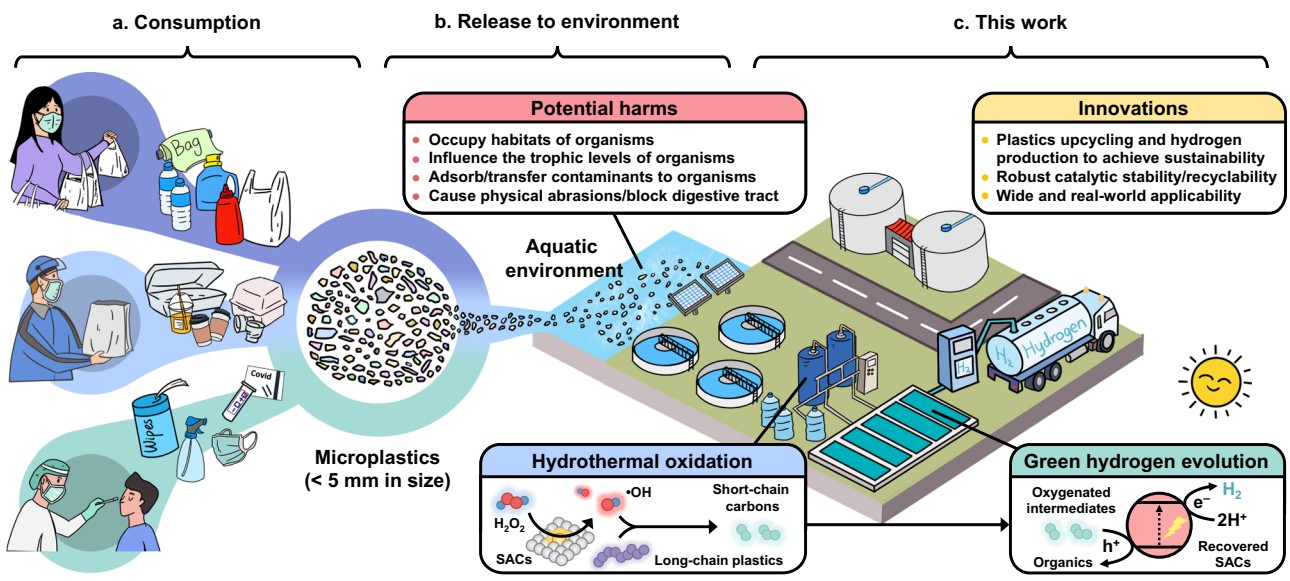

**Fig. 1 | Plastic consumption, release, and new strategy of microplastics upcycling. a** Consumption and generation of microplastic wastes. **b** Release and potential harms of microplastic wastes. **c**, Proposed strategy of a tandem microplastic degradation-hydrogen evolution reaction (MPD-HER) process.

are susceptible to the hydrolysis process, which limits the applicability of these approaches[11]. As another emerging strategy, Fenton/Fenton-like systems show promise by activating oxidants, like hydrogen peroxide ($H_2O_2$) or peroxymonosulfate (PMS), to produce reactive oxygen species (ROS) for MP degradation in hydrothermal conditions[14-15]. These ROS are effective in activating the C−H bond of plastics, applicable to degrade wide types of MPs[14-16]. However, current studies are limited by low MP degradation efficiency and poor catalyst recyclability in heterogeneous Fenton-like catalysis[14], and issues of harsh acid conditions (pH <3) and iron waste generation that necessitates post-treatment in homogeneous Fenton systems[8,15]. Moreover, the MP degradation mechanism and degradation intermediates/products were rarely investigated. To address the MP problem, more effective and sustainable innovations in both technology and catalysts are needed.

Very recently, single-atom catalysts (SACs) have revolutionized plastic upcycling, owing to their maximal metal-atom utilization and unique metal-support interaction[17,18]. For instance, Li et al.[18] reported a N-bridged Co/Ni dual-SAC for polystyrene (PS) conversion in a pressurized fixed-bed reactor. The dual Co/Ni sites ensured effective styrene molecules adsorption and C = C bond activation, achieving a 95 wt % PS conversion rate. Carbon nitride (CN), a well-known semiconductor photocatalyst and optimal support for SACs[19-21], provides abundant binding sites to immobilize single-atom metals and form metal−$N_x$ moieties[22,23], granting the catalyst robust reactivity and chemical stability[24]. To date, the application of CN-based SACs in MPs degradation and upcycling has not yet been explored.

This study introduces a tandem catalytic MPD-HER process (Fig. 1, bottom) by integrating MPs degradation and photocatalytic $H_2$ production using a hierarchical CN-supported single-atom Fe catalyst (FeSA-hCN). The FeSA-hCN with Fe−$N_4$ sites effectively activates $H_2O_2$ to generate hydroxyl radical (•OH) for breaking down ultrahigh molecular-weight polyethylene (UHMWPE). Almost complete UHMWPE degradation was achieved under neutral pH condition, with remarkable catalytic stability over 6 cycles (72 h). The effectiveness was demonstrated in degrading various everyday plastic products. This Fenton-like system, including MP degradation efficiency, catalyst stability, and mild pH condition, surpasses other systems in existing literature (Supplementary Table 1). Additionally, a high carboxylic acid product selectivity of 64% was achieved, which can induce a

subsequent solar-driven $H_2$ production process. The reaction mixture of FeSA-hCN and UHMWPE degradation products shows a $H_2$ evolution rate of 42 μmol h$^{-1}$, exceeding most reported photocatalytic plastic reforming systems (Supplementary Table 2). This work heralds a promising direction for developing effective catalysts for MP degradation and high-efficiency, selective upgrading, while also presenting a cost-effective method for clean fuel production and product utilization.

## Results
### Synthesis and characterizations of FeSA-hCN

The FeSA-hCN was synthesized using an elaborately designed silica template-confined molten salt strategy (Fig. 2a and Supplementary Fig. 1). Bulk CN and hCN were synthesized for comparison. The scanning electron microscopy (SEM) and transmission electron microscopy (TEM) images showed the periodically ordered and well-interconnected macroporous structure of FeSA-hCN, with abundant mesopores on the pore wall (Fig. 2b and Supplementary Fig. 2). The hierarchical porous structure (Supplementary Fig. 3 and Table 3) promotes active-site accessibility to reactants, giving rise to favorable catalytic kinetics[25,26]. The aberration-corrected high-angle annular dark field scanning transmission electron microscopy (AC HAADF-STEM, Fig. 2c and Supplementary Fig. 4) image of FeSA-hCN and line intensity profile indicated the uniform dispersion of isolated Fe single atoms within hCN matrix. X-ray diffraction patterns (XRD) showed the diffraction peaks for carbon nitride without crystalline Fe features, verifying the atomic dispersion of Fe (Supplementary Fig. 5). A high Fe loading (4.0 wt%) was determined by thermogravimetric analysis (TGA, Supplementary Fig. 6)[27].

C K-edge X-ray absorption near-edge structure (XANES) spectra (Fig. 2d) of CN and hCN featured two main characteristic peaks of π* C−C/C = C and π* C − N − C[28]. Two characteristic peaks at 399.5 and 402.4 eV in N K-edge spectra were assigned to π* C − N − C and π* N − C (i.e., C−N−2H and 2C−N−H), respectively[29]. After single-atom Fe loading, the intensity of intralayer π* N − C peak decreases, indicating the loss of H-bonding interactions and the successful incorporation of Fe atoms into the polymeric melon structure of carbon nitride[20]. Such structural features of FeSA-hCN were confirmed by XRD and Fourier transform infrared spectroscopy (FTIR) spectra (Supplementary Figs. 5, 7 and 8).

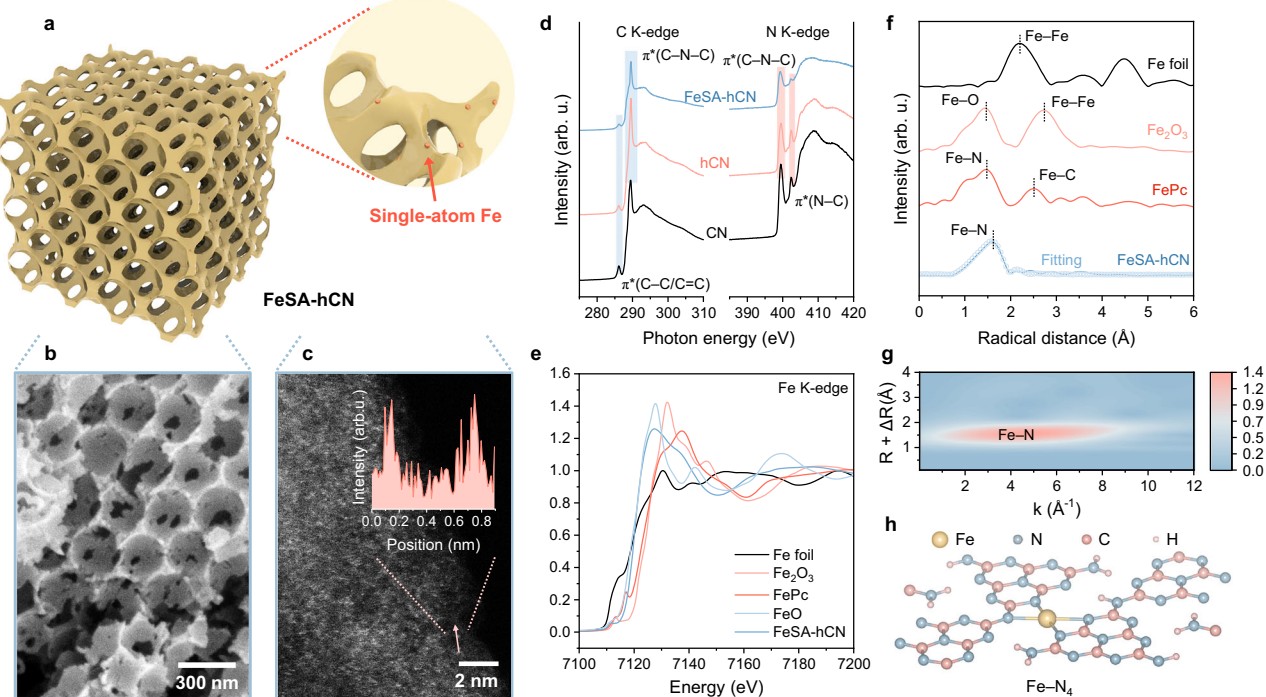

**Fig. 2 | Synthesis and structural characterizations of catalysts. a** Schematic illustration of FeSA-hCN. **b** SEM image of FeSA-hCN. **c** Aberration-corrected high-angle annular dark field scanning transmission electron microscopy (AC HAADF-STEM) image of FeSA-hCN and the corresponding line intensity profile along the red arrow (inset of Fig. 2c). **d** C and N K-edge XANES of FeSA-hCN, hCN, and CN. arb. u., arbitrary units. **e** Normalized Fe K-edge XANES of FeSA-hCN, Fe foil, FeO, Fe$_2$O$_3$, and FePc, with the average oxidation state of Fe in FeSA-hCN very close to FeO (+ 2). **f** $k^2$-weighted Fourier transform (FT) of the EXAFS spectra of FeSA-hCN, Fe foil, Fe$_2$O$_3$, and FePc. EXAFS fitted result of FeSA-hCN in R-space was presented in scatter type. **g** Wavelet transform (WT) EXAFS plot of FeSA-hCN. **h**, The theoretical model of Fe-C$_3$N$_4$ for the representation of FeSA-hCN according to the Fe K-edge EXAFS analysis result.

The chemical states and local coordination configuration of single-atom Fe in FeSA-hCN were further examined. The L-edge XANES spectra (Supplementary Fig. 9) evidenced that Fe mainly existed as Fe$^{2+}$ (Supplementary Table 4)[30]. The Fe K-edge XANES and corresponding first derivatives further confirmed that the oxidation state of Fe in FeSA-hCN is close to 2 (Fig. 2e and Supplementary Fig. 10). The partial oxidation state of Fe is favorable for H$_2$O$_2$ activation via Eq. 1 to generate •OH for MP oxidation (Eq. 1)[19,31].

$$Fe^{2+} + H_2O_2 \rightarrow Fe^{3+} + \bullet OH + OH^- \quad (1)$$

Fourier transform (FT) extended X-ray absorption fine structure spectroscopy (EXAFS) spectra displayed a dominant peak near 1.6 Å, ascribed to the Fe−N(C) first-shell coordination (Fig. 2f)[32]. No Fe−Fe interactions (2.2 Å) were observed, confirming the atomic dispersion feature of Fe atoms in FeSA-hCN. The quantitative EXAFS fitting curves (Fig. 2f and Supplementary Fig. 11) and the corresponding fitting results (Supplementary Table 5) revealed a Fe−N$_4$ coordination with an average bond distance of 2.1 Å. The wavelet transform (WT) contour plot further demonstrates the first-shell Fe−N pattern (Fig. 2g). Considering the possible metal−N(C) first-shell coordination in carbon nitride carriers[32,33], we established and optimized several theoretical structure models by density functional theory (DFT) calculations (Supplementary Table 6). Three types of N (N$_a$, N$_b$, N$_c$, Supplementary Fig. 12) exist. The Fe−N$_4$ structure by coordinating the Fe atom with three triangular edge N ($sp^2$-N$_b$) and one amino group/bridging N ($sp^3$-N$_a$) was thermodynamically favorable to form (Fig. 2h, Supplementary Fig. 12), which is consistent with the experimental results. DFT simulation also demonstrated that the Fe−N$_4$ sites could modulate the electronic structure of carbon nitride through charge redistribution (Supplementary Fig. 13), affecting its catalytic performance.

## UHMWPE MPs degradation by Fenton-like reaction

The plastic degradation tests were performed under hydrothermal conditions, employing H$_2$O$_2$ as the oxidant and UHMWPE MPs (melting point of 144 °C) as the target pollutant. With H$_2$O$_2$ alone, a 31.4 ± 1.3 wt% UHMWPE weight loss was observed after a 12 h treatment at 140 °C (Fig. 3a and Supplementary Fig. 14), and negligible enhancement was obtained when adding pristine CN or hCN. In contrast, FeSA-hCN exhibited a robust reactivity with an UHMWPE weight loss reaching 80.9 ± 5.3% (Fig. 3a and Supplementary Fig. 15). The gradual enhancement in degradation efficiency with elevating Fe loadings (0.5, 2.2 to 4.0 wt%) confirmed the significance of the Fe single-atoms (Supplementary discussion and Supplementary Figs. 16-20). The minimal catalyst weight loss of <5% (140 °C and 12 h) suggests the robust hydrothermal stability of FeSA-hCN. For subsequent experiments, the FeSA-hCN with optimum 4.0 wt% Fe was used for MP degradation unless otherwise specified. Control experiments were conducted, including UHMWPE MP degradation in pure H$_2$O, FeSA-hCN only, and H$_2$O$_2$ only systems (Supplementary Fig. 21).

The effects of pH values, H$_2$O$_2$ dosage, and catalyst dosage were investigated. As shown in Fig. 3b, FeSA-hCN exhibited stable catalytic activity in a wide pH range from 3 to 9. Increasing the pH to 11 may cause rapid decomposition of H$_2$O$_2$, hindering the reaction[34]. The mild pH operation conditions enhance the system's applicability in real-world settings. The optimum H$_2$O$_2$ dosage was determined as 100 mM. The addition of excess H$_2$O$_2$ (200 mM) might cause •OH scavenging and generate less reactive •OOH (Eq. (2))[31]. The catalyst dosage was also optimized (Supplementary Fig. 22).

$$H_2O_2 + \bullet OH \rightarrow H_2O + \bullet OOH \quad (2)$$

Under the optimized reaction conditions (neutral pH, 100 mM H$_2$O$_2$, and 1 g L$^{-1}$ catalyst) at 140 °C, MP degradation reached

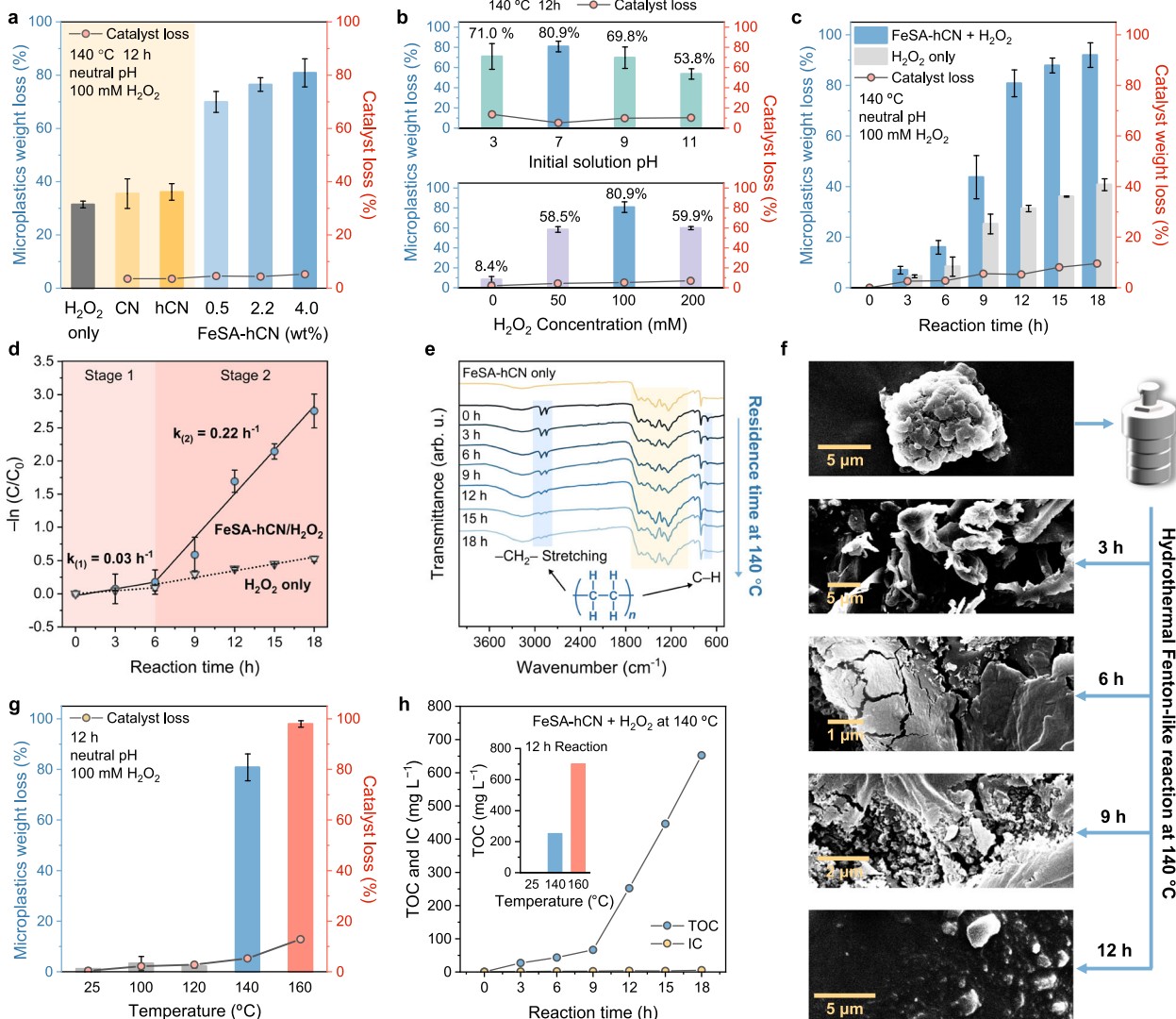

**Fig. 3 | Hydrothermal degradation of UHMWPE MPs in Fenton-like reaction.**
**a** MP and catalyst weight losses using CN, hCN, and FeSA-hCN. The MP weight losses were calculated after compensating the weight loss of catalysts. The catalysts weight loss partially arose from inevitable weighing and operational errors. **b** Effect of pH and $H_2O_2$ dosage on the weight loss of MPs and catalyst. **c** UHMWPE degradation performance in FeSA-hCN/$H_2O_2$ system and control system using $H_2O_2$ alone. **d** First-order kinetic curve fitting. **e** FT-IR spectra of FeSA-hCN and FeSA-hCN/UHMWPE reaction residues after 140 °C hydrothermal reaction at different resistance time. Marked bands (yellow) in 1000 to 1800 cm⁻¹ correspond to C − N heterocycles in carbon nitride (as compared to the pure UHMWPE in Supplementary Fig. 23). Marked bands (blue) correspond to the −CH₂− stretching and rocking deformation C−H of UHMWPE. **f** SEM images of fresh MPs and reaction residues after different reaction time (3, 6, 9 and 12 h). **g** Effect of hydrothermal reaction temperature on UHMWPE degradation performance and catalyst weight loss. **h** TOC and IC concentrations of reaction solution with different reaction time. TOC concentration of reaction solution after performance test under different reaction temperature (inset of Fig. 3h). Reaction conditions: [UHMWPE MPs] = 1 g L⁻¹, [catalysts] = 1 g L⁻¹, [$H_2O_2$] = 100 mmol L⁻¹, hydrothermal temperature = 140 °C, neutral pH, if not otherwise specified. The error bars represent the standard deviations, obtained by repeating the experiment twice.

92.0 ± 4.8% in 18 h (Fig. 3c). A two-stage UHMWPE degradation process was observed (Fig. 3d). The pseudo-first-order kinetic model showed the rate constant ($k$) of 0.03 and 0.22 h⁻¹ in 0-6 h and 6-18 h, respectively, which were 2 and 6.5 times higher than $H_2O_2$ alone. FTIR spectra in Fig. 3e and Supplementary Fig. 23 showed three main peaks at around 718, 2847, and 2915 cm⁻¹, corresponding to the vibrations of rocking deformation C−H, symmetric −CH₂− stretching, and asymmetric −CH₂− stretching groups of UHMWPE, respectively. The reduced intensity with prolonged reaction time confirmed an effective bond fracture in the UHMWPE chain (Fig. 3e). Morphological changes of UHMWPE degradation residues in FeSA-hCN/$H_2O_2$ system at 140 °C were shown in Fig. 3f: (i) within the initial 3 h, UHMWPE MPs aggregated into larger clumps; (ii) by the 6-h mark, UHMWPE pieces were cracked and fragmented; (iii) after 9-h reaction, the fragmentation

degree was enhanced with newly formed cracks; (iv) by 12 h, most UHMWPE MPs were degraded and only smaller particle residues were collected.

Morphologies of UHMPWE were also monitored in different control systems of pure water, $H_2O_2$, and CN/$H_2O_2$ at 140 °C. UHMPWE aggregated with time in these systems (Supplementary Figs. 24-26) with smooth surfaces, but some cracks occurred at 9 and 12 h in $H_2O_2$ and CN/$H_2O_2$, whereas bulk UHMPWE remained and cannot be effectively degraded as in FeSA-hCN/$H_2O_2$. The melting states of UHMPWE were then examined by observing the appearance changes of UHMWPE in pure water (Supplementary Fig. 27). UHMWPE MP powders aggregated into small fragments in 3 h, suggesting that they just started to melt. The fragments coalesced into a larger piece at 6 h, suggesting that more UHMPWE was melted. At 9 and 12 h, plastics

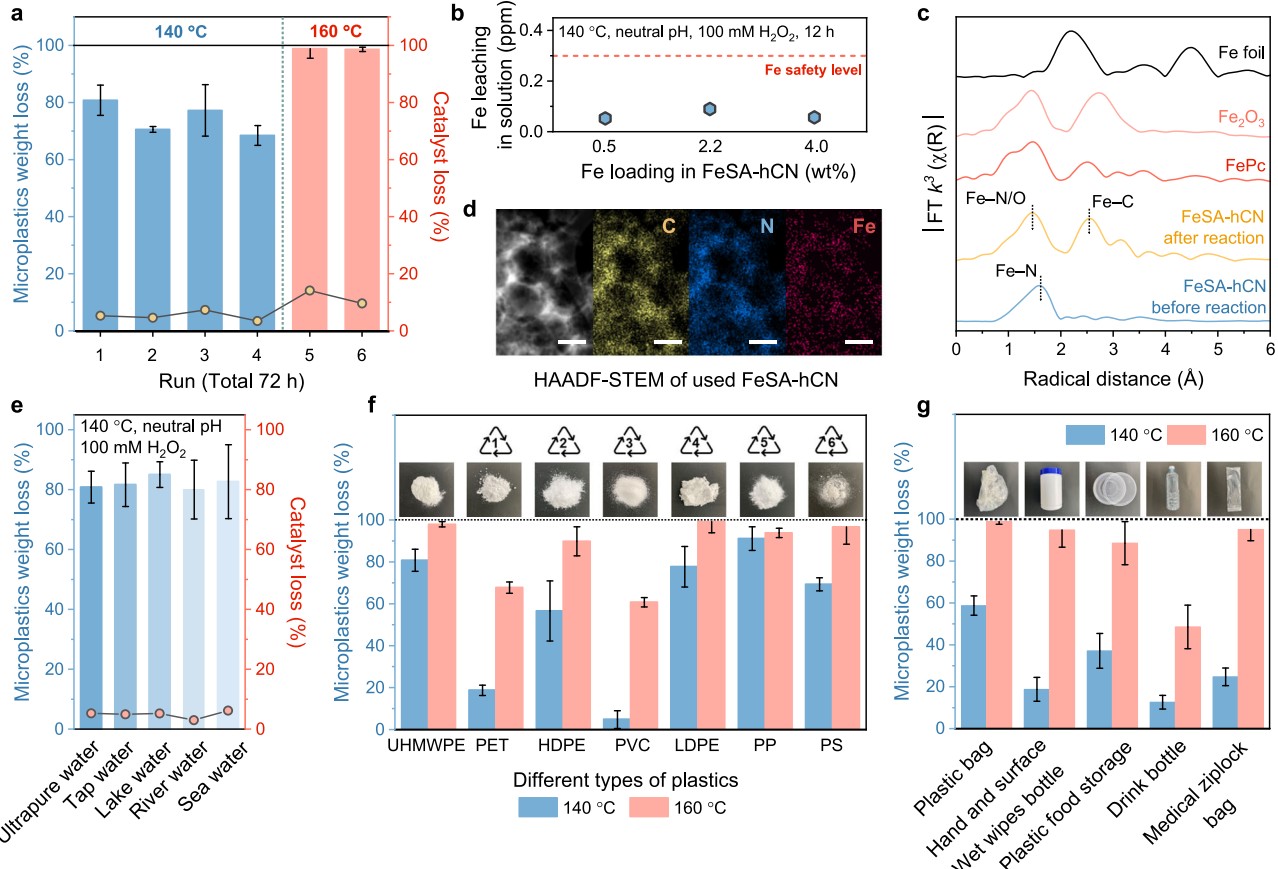

**Fig. 4 | Recyclability, stability, and applicability evaluation. a** Stability tests of FeSA-hCN: The first 4 runs were performed under 140 °C for 12 h and the last 2 runs were performed under 160 °C for 12 h. **b** Fe concentrations in solution. **c** FT spectra from Fe K-edge EXAFS in R space of used FeSA-hCN. **d** HAADF-STEM image and energy-dispersive X-ray spectroscopy (EDX) elemental mapping images of used FeSA-hCN, Scale bars, 200 nm. **e** Performance evaluation in different water matrices. **f** Performance evaluation using different plastics: PET, HDPE, PVC, LDPE, PP, and PS. **g** Performance evaluation using different types of real-life plastics. Reaction conditions: [MPs] = 1 g L$^{-1}$, [catalyst] = 1 g L$^{-1}$, [H$_2$O$_2$] = 100 mmol L$^{-1}$, hydrothermal temperature = 140/160 °C, and neutral pH if not otherwise specified. The error bars represent the standard deviations, obtained by repeating the experiment twice.

formed a more uniform sheet with a smooth surface (Supplementary Fig. 24), indicating a further or full melting state under hydrothermal conditions. We speculate that the two-stage UHMWPE degradation kinetics in Fig. 3d is related to the melting states and the degradation is slow when UHMWPE is not adequately melted in 3 and 6 h, while the kinetics will be accelerated after 6 h when UHMWPE MPs are mostly melted.

For the temperature effect, because UHMWPE degradation can only be achieved when the hydrothermal temperature approaches its melting point (144 °C)[15], a negligible UHMWPE weight loss was observed at 25, 100, and 120 °C (Fig. 3g). Increasing the reaction temperature from 140 to 160 °C can accelerate the melting of UHMWPE and facilitate radical generation for UHMWPE degradation thermodynamically[35]. UHMWPE weight loss (98 ± 2%) was achieved after a 12-h reaction at 160 °C, with no UHMWPE characteristic peak being observed in the corresponding FT-IR spectrum (Supplementary Fig. 28).

Different from previous homogenous Fenton processes that reported a high mineralization conversion of MPs into CO$_2$ and H$_2$O[15,36,37], total organic carbon (TOC) tests in FeSA-hCN/H$_2$O$_2$ system show that the TOC concentration in the reaction mixture slightly increased to 66.6 mg L$^{-1}$ in the first 9 h and surged to 652.8 mg L$^{-1}$ with the prolonged reaction time to 18 h at 140 °C (Fig. 3h and Supplementary Fig. 29). A higher TOC concentration (700.5 mg L$^{-1}$) was observed after a 12-h reaction at 160 °C. This growth trend was consistent with the weight loss results of MPs, suggesting that UHMWPE MPs were converted into liquid organic carbon species. We also tried

to detect gas products. Only trace CO$_2$ and H$_2$ were detected after the reaction, and their amounts were far lower than the organic chemicals in the solution (Supplementary Fig. 30). Signals of acetylene (C$_2$H$_2$) and methane (CH$_4$) were also found; however, their intensities were below the analytical limit (Supplementary Fig. 31). TOC and gas product analysis further verified that the UHMWPE MPs were mostly converted into organic chemicals.

## Recyclability, stability, and applicability evaluation

To assess the recyclability and chemical stability, the catalyst was tested for 6 consecutive cycles (72 h) (Fig. 4a and Supplementary Fig. 32). Minor fluctuations were observed in reactivity after 4 cycles at 140 °C. The recovered catalysts after the 4th run were further tested for UHMWPE degradation at 160 °C, showing almost complete MP removal efficiency in the 5th and 6th cycling runs. This demonstrated the robust catalytic stability of FeSA-hCN, whether reused continuously or alternately at 140 and/or 160 °C. The XRD, FTIR, and C/N K-edge XANES spectra showed no visible change of FeSA-hCN following the reaction (Supplementary Figs. 33-35), indicating its structural integrity under hydrothermal conditions. Only 0.1 ppm Fe leaching (12 h, 140 °C) was quantified by inductively coupled plasma mass spectrometry (ICP-MS, Fig. 4b). This was a trace level concerning the initial Fe loading throughout all FeSA-hCN catalysts (Supplementary Fig. 36). Importantly, this detected Fe concentration is well below the global maximum contamination for Fe in drinking water (0.3 ppm), as set by the World Health Organization (Fig. 4b)[38].

The FT EXAFS of used FeSA-hCN exhibited a first-coordination shell at 1.5 Å (Fig. 4c), corresponding to Fe−O and/or Fe−N coordinations. Compared to fresh FeSA-hCN, the emergence of Fe−O coordination resulted from the interaction between Fe sites and O atoms from the adsorbed $H_2O_2$ and its dissociative intermediates. An additional coordination shell at 2.5 Å was ascribed to the Fe−C interactions[39], which may originate from the interaction of Fe sites with UHMWPE polymer chains. Notably, the absence of a Fe−Fe peak around 2.2 Å confirmed the maintenance of the single-atom Fe site structure after hydrothermal reactions. This agrees well with the HAADF-STEM characterizations (Fig. 4d and Supplementary Fig. 37).

The FeSA-hCN/$H_2O_2$ system also exhibited excellent real-life applicability in oxidizing MPs in different natural water bodies, including tap, lake, river, and seawater (Fig. 4e). We then investigated the degradation performance of FeSA-hCN on different polymer types (Fig. 4f and size information is presented in Supplementary Fig. 38), namely, polyethylene terephthalate (PET), high-density polyethylene (HDPE), polyvinyl chloride (PVC), low-density polyethylene (LDPE), polypropylene (PP) and polystyrene (PS). The weight losses of UHHDPE, LDPE, PP, and PS reached >70% at 140 °C and >90% at 160 °C within a 12-h reaction. PET and PVC were relatively resistant to degradation at 140 °C, due to their higher melting points ( > 160 °C)[15]. Nevertheless, they started to decompose when the reaction temperature increased to 160 °C. Further expanding the plastic substrates to some daily life plastics, like plastic bags, 'hand and surface wet wipes' plastic bottles, plastic food containers, plastic drink bottles, and medical ziplock bags, the catalytic system maintained its high degradation performance (Fig. 4g and Supplementary Fig. 39). Over 90% degradation efficiencies were achieved in most real-life plastic products at 160 °C after 12 h. The high activity, stability, and applicability of this system in these actual conditions endow it with excellent practical application prospects, offering unique potential in addressing real-life environmental plastic pollution.

## Mechanism study

To date, different Fenton-like systems have been reported, including persulfate-based AOP[14], thermal-assisted Fenton reaction[15,37], photo-Fenton reaction[40], and electro-Fenton-like reaction[16] to degrade MPs. The degradation mechanisms vary based on different configurations of catalysts, oxidants, irradiation, and electrochemistry, producing different ROS (e.g., sulfate radical, superoxide radical ($O_2^{\cdot-}$), •OH) or involving photo-induced holes or electron transfer[8]. To explore the ROS involved in FeSA-hCN/$H_2O_2$ system, we conducted electron paramagnetic resonance (EPR) and chemical quenching experiments (Fig. 5a and b, Supplementary Figs. 40 and 41). The EPR analysis showed that FeSA-hCN/$H_2O_2$/DMPO system produced a 4-fold characteristic peak with an intensity ratio of 1:2:2:1, corresponding to •OH (Fig. 5a). No $O_2^{\cdot-}$ was detected (Supplementary Fig. 40). The quenching experiment confirmed the significant contribution of •OH on UHMWPE degradation (Fig. 5b and Supplementary Fig. 41). The $H_2O_2$ consumption test evidenced that $H_2O_2$ has been effectively activated (Supplementary Figs. 42 and 43). While the consumption rate of $H_2O_2$ was fast in the first 3 h, the UHMWPE degradation rate was low in the first 6 h (Fig. 4c, d), which should be closely related to the melting states of MPs. As above-mentioned, UHMWPE was aggregated and was not adequately melted in 0-6 h, making it difficult for the as-produced •OH to attack. After 6 h, UHMWPE was mostly melted, making it easier to be oxidized by •OH. $H_2O_2$ was detected to be at 0.05 mM after 18 h at 140 °C, indicating that the amount of $H_2O_2$ was sufficient to support the production of •OH during the whole reaction time.

The Fe K-edge XANES spectrum shows a positive shift in FeSA-hCN after the reaction, indicating an increased Fe valence state (Fig. 5c). Fe L-edge spectrum further proved the predominance of $Fe^{3+}$ and a minor presence of $Fe^{2+}$ (Supplementary Fig. 44 and Table 4). $Fe^{2+}$ in FeSA-hCN was transformed into $Fe^{3+}$ while activating $H_2O_2$ to

produce •OH for MPs degradation (Eq. (1))[41]. Upon reuse, the as-produced $Fe^{3+}$ could react with $H_2O_2$ to regenerate $Fe^{2+}$ (Eq. (3))[19]. The redox $Fe^{2+}$/$Fe^{3+}$ cycles endow FeSA-hCN with excellent reusability and stability for efficient $H_2O_2$ activation.

$$Fe^{3+} + H_2O_2 \rightarrow Fe^{2+} + \bullet OOH \qquad (3)$$

Further, we verified the advantage of using Fe-hCN in $H_2O_2$ activation through theoretical calculations, using Fe-$C_3N_4$ as a theoretical model to represent FeSA-hCN. As shown in Fig. 5d, an $H_2O_2$ molecule is more favorable to be adsorbed on Fe-$C_3N_4$ than $C_3N_4$. The obviously stretched O−O bond of $H_2O_2$ on Fe-$C_3N_4$ (1.46 Å to 2.37 Å) indicated that $H_2O_2$ was efficiently activated on Fe-$C_3N_4$, making it easier to break up and form •OH[42]. The free energy profile demonstrates that the dissociation of $H_2O_2$ into surface hydroxyl species (*$H_2O_2$ → *OH + *OH) on the Fe site of Fe-$C_3N_4$ is thermodynamically favorable (Fig. 5e and Supplementary Fig. 45)[43]. By contrast, the $H_2O_2$ dissociation on $C_3N_4$ confronted a large activation barrier of 1.70 eV, making it the rate-determining step (RDS). For Fe-$C_3N_4$, the RDS is the formation of •OH + *OH, with a relatively low free energy barrier of 1.62 eV. Spatial charge density distribution and Bader charge analysis revealed that $H_2O_2$ tended to obtain much more charges from Fe-$C_3N_4$ (0.76 e) than from $C_3N_4$ (0.05 e) (Fig. 5d and Supplementary Fig. 46). The total and projected DOS of Fe-$C_3N_4$*$H_2O_2$ showed obvious Fe 3$d$ and O 2$p$ orbital contributions near the Fermi energy level with the distribution of O 2$p$ above the Fermi level (Fig. 5f and Supplementary Fig. 47). This suggests an apparent charge transfer from the Fe 3$d$-orbital of Fe-$C_3N_4$ to the O 2$p$-orbital of $H_2O_2$. The fast transfer rate between Fe single sites and $H_2O_2$ can impel the activation of $H_2O_2$[42]. In contrast, no noticeable change was observed on $C_3N_4$. From a theoretical perspective, the integration of obvious stretched O−O bond of $H_2O_2$, strong $H_2O_2$ adsorption, low energy barrier of RDS, and favorable electronic structure of Fe-$C_3N_4$ is responsible for the superior activity of FeSA-hCN in $H_2O_2$ activation to hCN or CN.

The reaction intermediates/products from UHMWPE degradation are identified by gas chromatography-mass spectrometry (GC-MS, Supplementary Figs. 48 and 49, Supplementary Tables 7 and 8). As shown in Fig. 5g and h, an induction period was observed in the early stage of hydrothermal reaction (3 h)[44], and the organic products rapidly accumulated from 3 to 18 h. Early reaction intermediates (retention time around 4 minutes) were identified as signals corresponding to the formation of ketone functional groups (C = O). Combined with the EPR and quenching experiments, it could be concluded that the hydrocarbon chains of UHMWPE were partially oxidized (C−H bond activation) by •OH in the induction period (Fig. 5i)[44]. This was verified by the FTIR analysis of UHMWPE collected after the reaction. C = O and C − O groups appeared and gradually increased with the prolonged reaction time (Supplementary Fig. 23). After 6 h, multiple organic products (e.g., carboxylic acid, ether, alkane, furanone, etc.) were detected (Fig. 5H and Supplementary Table 7), which are within the gasoline and diesel range ($C_3$-$C_{20}$)[45]. Significantly, the selectivity of carboxylic acids (R − COOH) reached >60% (mainly in the $C_3$-$C_{10}$ range (Supplementary Fig. 50)).

The partially oxidized UHMWPE was easily attacked by •OH[44] (Fig. 5i), leading to the breakage of C − C bond, scission of the polymer backbone, and the formation of the hydroxyl ( − OH) groups over its C = O containing carbon chain for carboxylic acid (R − COOH) production. In the time-evolution experiments at 140 °C, the product selectivity of carboxylic acids increased from 58% to 64% when the reaction time increased from 6 to 12 h but decreased to 61% at 18 h (Fig. 5h). Elevating the reaction temperature to 160 °C resulted in the formation of amide products, hindering the selective production of carboxylic acids. The ecotoxicities (including acute and chronic toxicities to fish, daphnids, and green algae) of possible organic products were evaluated using the Ecological Structure Activity Relationships 19

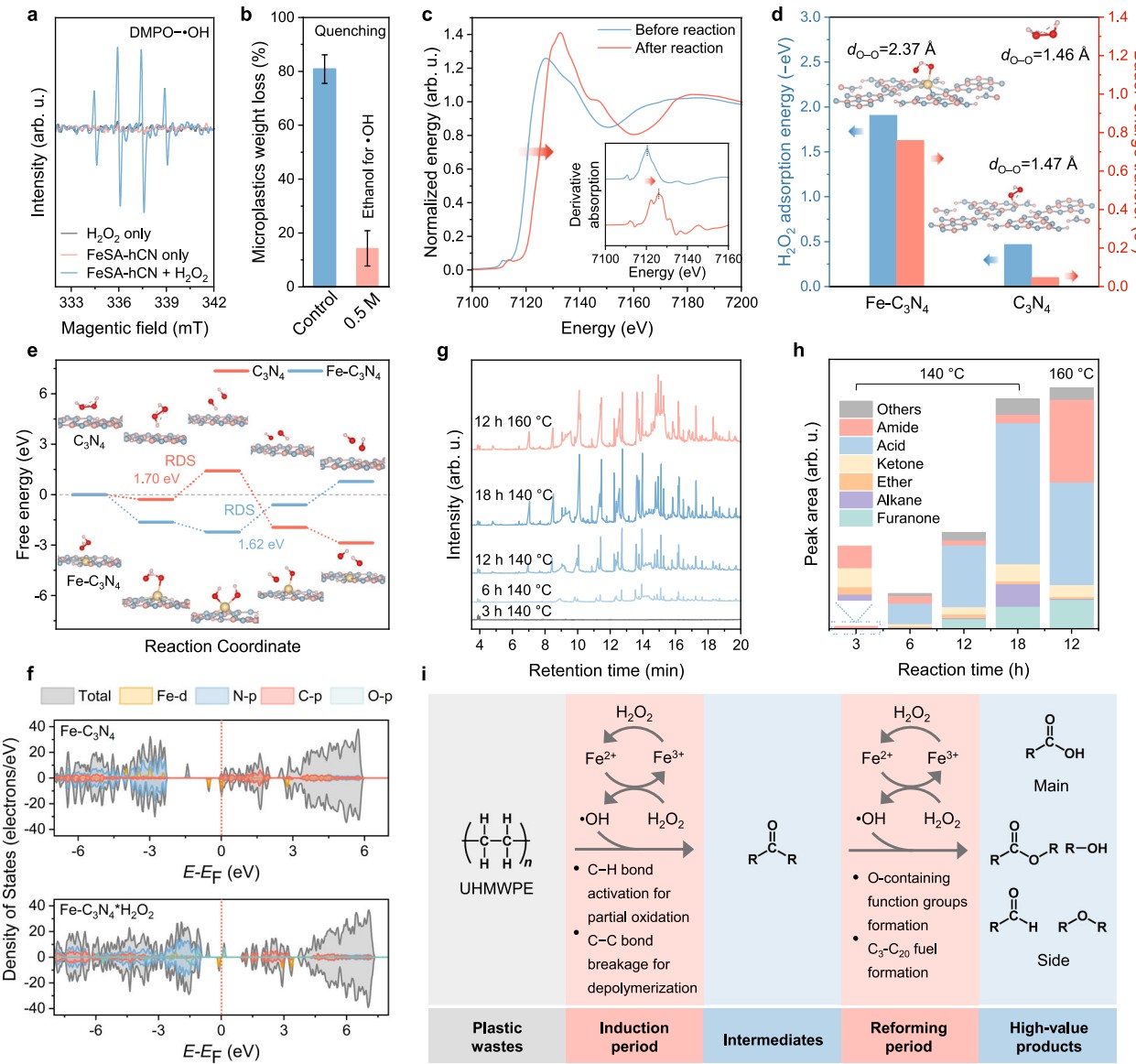

**Fig. 5 | Reaction mechanism study and product identification. a** EPR spectra for FeSA-hCN catalyzed Fenton-like reactions. **b** Effect of the quenching agent (ethanol) on UHMWPE degradation for the FeSA-hCN catalyst. The error bars represent the standard deviations, obtained by repeating the experiment twice. **c**, Fe K-edge XANES of used FeSA-hCN after UHMWPE degradation test. First derivatives of Fe K-edge XANES (inset of Fig. 5c). **d** Theoretical study of $H_2O_2$ adsorption, O−O bond length ($d_{O−O}$), and Bader charge transfer of Fe-$C_3N_4*H_2O_2$ and $C_3N_4*H_2O_2$. **e** Energy profiles for the $H_2O_2$ activation process. **f** Total and projected DOS of Fe-$C_3N_4$ and Fe-$C_3N_4*H_2O_2$. **g** Gas chromatography-mass spectrometry (GC-MS) analysis of reaction products after different residence time. **h** Reaction products investigation. **i**, UHMWPE degradation mechanism.

(ECOSAR) Software (Supplementary Figs. 51 and 52). The majority of reaction intermediates and products showed low acute and chronic toxicities to fish, daphnids, and green algae. Our FeSA-hCN/$H_2O_2$ Fenton-like system excels in MP degradation efficiency, catalyst stability, versatility, and mild pH operation conditions, setting a new benchmark beyond the currently documented literature (Supplementary Table 1). Additionally, we address and fill the knowledge gap in the analysis of plastic degradation intermediates/products, offering new insights into the potential use of degraded plastic components. Future studies will be made toward further optimizing the process's efficiency and selectivity, exploring its practical viability in large-scale applications.

## Photocatalytic $H_2$ production using the degradation mixture
Following the examination of plastic degradation products, we now assess their practical value in solar fuel production. The predominant presence of carboxylic acids (product selectivity of 64%) in the UHMWPE degradation products has led to their consideration as potential sacrificial agents for sustainable photocatalytic hydrogen production (Fig. 6a). In such a process, carboxylates bind to the photocatalyst surface and proceed decarboxylation by reacting with photogenerated holes, boosting $H_2$ production from proton reduction[46–48].

Specifically, the reaction mixture of FeSA-hCN/$H_2O_2$/UHMWPE after the Fenton-like reaction was directly used for photocatalytic $H_2$ production. FeSA-hCN now works as a photocatalyst (Supplementary Fig. 53). The reaction solution withdrew after 12 h reaction at 140 °C exhibited the optimal $H_2$ evolution rate of 42 µmol h$^{-1}$ under simulated solar illumination (Fig. 6b). Compared to pure water, the dissociative electronegative −COOH group in the reaction mixture was adsorbed on FeSA-hCN's surface, and reacted with the photoinduced holes, thus promoting the separation of photogenerated carriers and improving

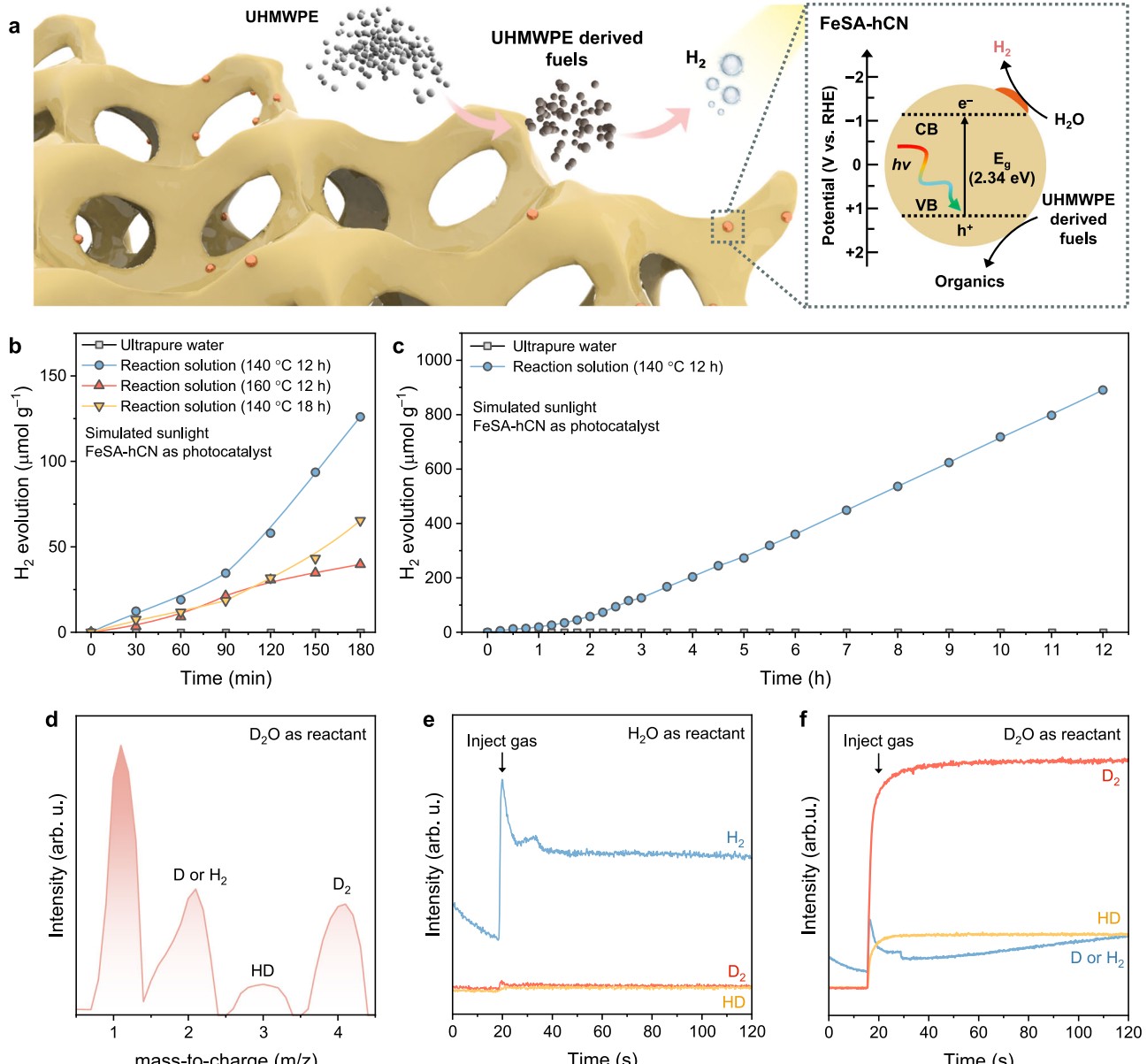

**Fig. 6 | Photocatalytic hydrogen production using reaction product mixture.**
**a** Illustration of photocatalytic hydrogen production mechanism using the reaction product mixture after hydrothermal degradation of UHMWPE MPs and the band structure of FeSA-hCN, determined based on Supplementary Fig. 53. **b** Photocatalytic hydrogen production performance using reaction product mixture under different reaction conditions under simulated AM1.5 G solar illumination (100 mW cm$^{-2}$) **c** Long-term photocatalytic hydrogen production test. **d** Online isotope ratio MS spectra. **e**, MS spectra of gas products using $H_2O$ as the reactant. **f** MS spectra of gas products using $D_2O$ as the reactant.

photocatalytic $H_2$ evolution[48,49]. The reaction solution after longer reaction time or higher temperature showed a lower mass ratio of carboxylic acid in the product mixture, thus decreasing the $H_2$ evolution rate (Fig. 6b). For the reaction mixture after 12 h reaction at 140 °C, the long-term photocatalytic test indicated that carboxylic acid is in excess for use as a sacrificial agent (Fig. 6c). The fundamental mechanism associated with UHMWPE-derived sacrificial reagents is whether they serve as the hole scavenger or donate their hydrogen atoms to produce $H_2$. To determine the origin of $H_2$, we did isotopic experiments with deuterium oxide ($D_2O$, 99.9% D) and $H_2O$ as reactants. The mass-to-charge (m/z) isotopic profile of $H_2$/D (m/z 2), H−D (m/z 3), and D−D (m/z 4) can be analyzed using online MS spectra (Fig. 6d). Only $H_2$ was detected using $H_2O$ as the substrate (Fig. 6e). In comparison, $D_2$ was detected as the main product using $D_2O$ as the substrate (Fig. 6f). This confirmed that the evolved $H_2$ originates

mainly from water, while UHMWPE-derived organics serve as the sacrificial reagents. Compared to existing methods for hydrogen production from plastic reforming (Supplementary Table 2), our approach avoids the use of strong acids or bases for plastic hydrolysis and expands the variety of plastics that can be processed in real water. Compared to photocatalytic methods, we significantly enhance hydrogen production efficiency.

## Discussion
In summary, we present a catalytic tandem MPD-HER process for sustainable plastic management and renewable hydrogen production using a FeSA-hCN catalyst. The Fe−N$_4$ site in FeSA-hCN can effectively activate $H_2O_2$ to produce •OH for UHMWPE decomposition, leading to the breakage of the C−C bond, the introduction of ketone functional groups (C=O) in the induction period, and the formation of the

hydroxyl ($-OH$) groups over its $C=O$ containing carbon chain for carboxylic acid ($R-COOH$) production. At neutral pH conditions, an ultrahigh UHMWPE degradation efficiency could be reached at $92.0 \pm 4.8\%$ after 18 h, 140 °C and $98.0 \pm 1.3\%$ after 12 h, 160 °C. Moreover, the FeSA-hCN shows robust stability with no reactivity decline after 6 cycles (over 72 h). Remarkably, excellent catalytic efficiency can be maintained in actual water bodies and daily-life single-use plastics, such as bags, drinking bottles, and takeaway boxes. Carboxylic acid selectivity (64%) was achieved in the plastic degradation product. Under illumination, the remaining FeSA-hCN in the degradation system can induce tandem photocatalytic $H_2$ production with a $H_2$ yield of 42 μmol h$^{-1}$, with the organic UHMWPE degradation products as the sacrificial reagents and water as the H source. This work demonstrates the effective degradation of real-world plastic waste and exemplifies the potential of utilizing plastic conversion products for hydrogen production, offering a promising pathway for the future of plastic waste management and energy production. Future research will focus on overcoming practical application barriers, broadening the range of degradable plastics, and further optimizing the process's efficiency and selectivity.

## Methods

### Materials
All the chemicals in this work are of an analytical grade without further purification. Ammonium hydroxide solution ($NH_3 \cdot H_2O$, 28.0 - 30.0% $NH_3$ basis), ammonium hydrogen difluoride ($NH_4HF_2$), acetone ($\geq 99.5\%$), Deuterium oxide ($D_2O$, 99.9%), iron(III) acetylacetonate ($Fe(C_5H_7O_2)_3$), dicyandiamide (DCD), ethanol (absolute for analysis), hydrochloric acid (37% (w/w), HCl), hydrogen peroxide solution ($H_2O_2$, 30 wt%), tetraethyl orthosilicate (TEOS, ≥99.0%), ultra-high molecular weight polyethylene (UHMWPE powder, 0.94 g mL$^{-1}$ average Mw 3,000,000–6,000,000), poly(ethylene terephthalate) (PET, granule), high-density polyethylene (HDPE) pellets, polyvinyl chloride (PVC, powder, low molecular weight), low-density polyethylene (LDPE, 0.925 g mL$^{-1}$) pellets, polypropylene (PP, average Mw -12,000) pellets, and polystyrene (PS, average Mw 35,000) pellets were purchased from Sigma-Aldrich. Ultra-pure water (18.2 MΩ cm$^{-1}$) was used. Tap water was collected without going through any water purifier. Seawater was acquired on the beach of Grange in Adelaide, South Australia. Lake water was collected from the Waterford Lake in Northgate, South Australia. River water was collected from the River Torrens, South Australia. All the practical water samples were filtered using a 0.45 μm cellulose acetate (CA) membrane filter. The plastic bags, plastic food storage, and plastic bottles were obtained from Woolworths supermarket in Adelaide, South Australia. Hand and surface wet wipes bottles were purchased from Winc in Australia. COVID-19 antigen test (Nasal Swab) self-test pack was purchased from Chemist Warehouse in Adelaide, South Australia.

### Pretreatment of plastics
PET, HDPE, LDPE, and PP pellets were smashed by a grinder (IKA A 11 basic analytical mill, grinding for 10 s every 10 min). PS microplastics were prepared via ball-milling of PS pellets using a planetary mill (FRITSCH PULVERISETTE 7 with zirconia balls and vials) at a speed of 300 rpm overnight, operating for 2 min in every 7 min. The obtained particles were filtered by a 100 μm stainless steel sieve. Real-life plastic bags, food storage, drinking bottles, wet wipes bottles, and medical ziplock bags were cut into debris ($<5$ mm). After rinsing with ethanol and water, the plastics were dried in air for 3 days. The particle sizes of commercial UHMWPE, PET, HDPE, PVC, LDPE, PP, and PS powders were analyzed using the Mastersizer 2000 - Malvern.

### Synthesis of ordered silica template
Ordered silica microspheres ( - 300 nm) were fabricated via a modified Stöber method[50]. Solution A: 192 mL ethanol was added to a beaker

with 8 mL TEOS and stirred at 500 rpm under 25 °C. Solution B: 14 mL of $NH_3 \cdot H_2O$ was added to another beaker with 56.6 mL ethanol and 29.4 mL ultrapure water, and stirred at 500 rpm under 25 °C. After stirring for 30 min, solution B was slowly added to solution A at 800 rpm under 25 °C, and was kept stirring. Later, the as-obtained mixture was centrifuged at 18,900 g and 20 °C ($\pm 4$ °C). The collected silica solids were rinsed with ultrapure water to remove the unreacted residues. After drying in an oven at 60 °C overnight, white solids were then collected, ground and dispersed in 5 wt% ultrapure water. The as-obtained solution was dropped into vials (10 mL) for ultrasonication and evaporation at 110 °C for 24 h. The silica microspheres were self-assembled on the wall of the vials, forming silica strips.

### Synthesis of pristine carbon nitride (CN)
Pristine carbon nitride was prepared by directly calcining 0.6 g dicyandiamide at 520 °C for 2 h with a ramp of 2 °C·min$^{-1}$ and further to 550 °C for 2 with a ramp of 4 °C min$^{-1}$ h under a nitrogen gas atmosphere. The as-obtained solids were washed with ultrapure water and dried at 60 °C. Finally, the collected samples were labeled as CN.

### Synthesis of hierarchical porous carbon nitride (hCN)
Dicyandiamide (0.6 g) was dispersed on the silica template surface and calcined at 520 °C for 2 h with a ramp of 2 °C min$^{-1}$ and further to 550 °C for 2 h with a ramp of 4 °C min$^{-1}$ under a nitrogen gas atmosphere. The as-obtained product was etched with 50 mL $NH_4HF_2$ solution (4 M) for 48 h to remove the silica template completely. After that, the as-obtained solid was washed with ultrapure water several times and dried at 60 °C overnight. The collected sample was labeled as hCN.

### Synthesis of FeSA-hCN
In the process, 0.6 g dicyandiamide and iron(III) acetylacetonate (a certain amount) were dissolved in a 300 mL solution composed of water, ethanol, and acetone in a 1:1:1 ratio. Subsequently, the solution was stirred at 300 rpm under 70 °C for evaporation. The as-obtained solids were collected and ground for further use. Then, the well-mixed solids were uniformly tiled on the surface of 1.0 g silica template strips, calcined at 520 °C for 2 h with a ramp of 2 °C min$^{-1}$ and further to 550 °C for 2 h with a ramp of 4 °C min$^{-1}$ under $N_2$ atmosphere. The as-obtained product was etched with 50 mL $NH_4HF_2$ solution (4 M) for 48 h to remove the silica template and washed with 60 mL mixed water/HCl/ethanol solution (1:1:1) for 24 h to remove surface metallic species. Then, the solid was collected, washed with ultrapure water, and dried in an oven at 60 °C overnight.

### Electron microscopy
The morphologies of catalysts and microplastics were investigated by scanning electron microscopy (SEM, FEI Quanta 450). Transmission electron microscopy (TEM) images were obtained using FEI Tecnai G2 Spirit TEM. High-angle annular dark-field scanning TEM (HAADF-STEM) with energy-dispersive X-ray spectroscopy (EDX) elemental mapping images were acquired on an FEI Titan Themis 80–200.

### Spectroscopy
X-ray diffraction (XRD) was carried out using a Rigaku MiniFlex 600 X-ray diffractometer. Fourier transform infrared (FTIR) spectra were performed on Nicolet 6700 Thermofisher. X-ray absorption near-edge structure (XANES) spectra for C, N K-edge and Fe L-edge were obtained using the Soft X-ray Spectroscopy beamline at the Australian Synchrotron in Melbourne. All the spectra data were processed and analyzed using the QANT software program developed by the Australian Synchrotron[51]. The XANES of Fe K-edge and extended X-ray absorption fine structure (EXAFS) spectra were measured in a transmission mode at the Australian Synchrotron in Melbourne. Fe foil, FeO and $Fe_2O_3$ were regarded as the standard references. All the spectra were

collected under ambient conditions, data was analyzed using standard methods. The resulting spectra were energy calibrated, background corrected and normalized at the height of the edge step using the ATHENA module in the IFEFFIT packages[52].

### Determination of Fe loadings on FeSA-hCN

The iron loading content (weight percentage) was examined by a thermogravimetric analysis-differential scanning calorimeter (air, 800 °C, TGA–DSC, Mettler Toledo).

### Determination of Fe leaching into hydrothermal reaction solution

Fe leaching in the reaction solution was monitored by the inductively coupled plasma mass spectrometry (ICP-MS) using Solution ICP-MS-Agilent 8900x QQQ-ICP-MS.

### UHMWPE degradation evaluation

For the degradation, $1 g L^{-1}$ UHMWPE MPs and $1 g L^{-1}$ FeSA-hCN catalysts were first dispersed in a 70 mL Teflon autoclave with 50 mL ultrapure water. The total weights of UHMWPE and catalysts were determined by the Mettler Toledo analytical weighing scale. The as-obtained mixture was magnetically stirred at 1200 rpm for 25 min. After that, 100 mM $H_2O_2$ was added to the Teflon autoclave. Then, the Teflon autoclave was carefully sealed and rapidly transferred to a pre-warmed oven with a set temperature of 140 °C unless specifically mentioned. After the reaction, the autoclave was naturally cooled down to room temperature in the oven. The reaction residues were collected by vacuum filtration with 2 layers of pre-weighted 0.45 μm cellulose acetate (CA) membrane. Note that the catalysts and unreacted MPs on the autoclave wall were carefully removed and collected. The autoclave was rinsed several times with ultrapure water to ensure cleanliness. Then, the collected product (on CA membrane) was dried in air for 2 days before weighing. The catalyst weight loss was determined using a similar procedure without adding UHMWPE MPs.

### Cycling UHMWPE degradation tests

The dried reaction residues on the CA membrane were dispersed in a 70 mL beaker with 50 mL ultrapure water. After sitting for 24 h, most unreacted UHMWPE tended to float over water. These UHMWPE samples were carefully removed using tweezers. The remaining FeSA-hCN catalysts were collected by filtration, washed alternately with water and ethanol, and dried at 60 °C for the next cycling run.

### FTIR analysis of reaction residues

The dried reaction residues on the CA membrane were carefully transferred to an agate mortar, followed by grounding. The well-mixed solids were measured by a Nicolet 6700 FTIR Spectrometer (Thermofisher).

### FTIR analysis of UHMWPE after reaction

After reactions with different time, the UHMWPE was carefully removed from the Teflon reactor using tweezers. The UHMWPE was collected using a 500 mL beaker and washed with 300 mL water to remove catalysts on the UHMWPE surface. After magnetically stirring at 600 rpm for 2 h, the UHMWPE was collected by vacuum filtration. After drying, the UHMWPE was analyzed using the Nicolet 6700 FTIR Spectrometer (Thermofisher).

### Photocatalytic $H_2$ production evaluation

The photocatalytic $H_2$ production tests were conducted in a customized sealed black poly(tetrafluoroethylene) reactor with a quartz window. A 300 W xenon-arc lamp (with a standard AM 1.5 G filter, Aulight CEL-HX, Beijing) was used as a light source. A water circulation system was supplied to keep the temperature at 25 °C. After the hydrothermal reaction, the reaction solution was directly used for the

performance test without further treatment. Before the irradiations, $1 mg L^{-1}$ $H_2PtCl_6$ solution was added and vigorously stirred in the dark for 30 min. An ultrapure $N_2$ gas was used to purge the system at the same time to maintain the anaerobic conditions. The produced $H_2$ was analyzed by an online gas chromatograph (Agilent 490 Micro GC) using a thermal conductivity detector.

### Detection of gases produced during hydrothermal reaction

An externally heated autoclave reactor (Aulight CEL-MPR micro-reactor, Beijing) equipped with gas sampling ports was used. UHMWPE MPs ($1 g L^{-1}$) and $1 g L^{-1}$ FeSA-hCN catalysts were added into the reactor with 50 mL ultrapure water. After that, 100 mM $H_2O_2$ was added. Then, the autoclave reactor was carefully sealed, and the operation temperature was set at 140 °C for 18 h. After the reaction, the reactor was allowed to naturally cool down to room temperature. A gas sampling needle is then employed to extract a gas sample from the sampling port and inject it into the gas chromatography (Agilent 8890) for analysis.

### Electronic band structure study

UV-vis diffuse reflectance spectra were measured using a Cary 100 UV-vis spectrophotometer (Agilent, US). The bandgap of FeSA-hCN catalyst collected after hydrothermal UHMWPE degradation was estimated by the Tauc plots, obtained from the Kubelka-Munk function. The valence state position was determined by the valence band X-ray photoelectron spectroscopy (XPS, Thermo Scientific K-Alpha).

### Reaction products analysis using gas chromatography-mass spectrometry (GC-MS)

GC-MS analysis of the hydrothermal oxidation intermediates/products was completed using an Agilent 5977B MS coupled with a 7890B GC. A 30 m, 250 μm ID and 0.25 μm film thickness Agilent HP-5MS column was used for analysis with helium as the carrier gas at a flow of 1 mL/min. A sample (1 μL) was injected directly with an injection temperature of 300 °C. The temperature program starts with the initial temperature at 50 °C, held for 1 min, and then heated up to 300 °C with a ramping rate of 10 °C/min, and a final hold at 14 min. Mass spectrometric detection was operated with a scanning range of the m/z 50–500. An Agilent Chemstation software was used to undertake the data interpretation with NIST14 mass spectral library as the database. The acute and chronic toxicities of reaction intermediates and products were evaluated by the Ecological Structure Activity Relationships 19 (ECO-SAR) Software, developed by the United States Environmental Protection Agency.

### Online mass spectrometry (MS) analysis to determine the origin of $H_2$

The Hiden HPR-40 DSA Membrane Inlet Mass Spectrometer (MIMS) was used for the online mass spectrometry (MS) analysis. The gas produced from the photocatalytic $H_2$ production system was collected using a vacuum-cleaned gas bag. The gas bag was carefully pressed to allow gas to be drawn into the vacuum chamber for mass spectrometer analysis. To perform the isotopic experiments, deuterium oxide ($D_2O$, 99.9% D) and ultrapure water ($H_2O$) were used as reactants for hydrothermal UHMWPE MPs degradation experiments. After that, the reaction product mixture was used directly for photocatalytic $H_2$ production. The as-produced gas was collected for online MS analysis.

### Computational method

Density functional theory (DFT) calculations were performed by using the Vienna ab initio simulation package (VASP, 5.4.4)[53,54]. The Perdew-Burk-Ernzerhof (PBE) within the generalized gradient approximation (GGA) was used to describe the exchange-correlation functional[55]. The electron-ion potential was described by the projected augmented wave method (PAW)[56]. The plane wave energy cut-off is set to 500 eV.

The convergence criterion of the force was -0.03 eV/Å and the convergence threshold for the total energy was $10^{-5}$ eV. Since GGA cannot correctly describe strongly correlated systems containing partially filled d subshells, the GGA + U method is applied to describe partially filled $d$ orbitals by taking into account Coulomb and exchange corrections. We used a correlation energy (U) of 4 eV and an exchange energy (J) of 1 eV for Fe $3d$ orbitals. The Brillouin-zone integration was approximated $4 \times 3 \times 1$ by grid using the Monkhorst-Pack k-point mesh. A vacuum space of 15 Å in the Z-direction was chosen to avoid artificial interaction. The effect of van der Waals (vdW) interactions was included for weak interaction cases using the semiempirical correction scheme of Becke-Jonson damping, DFT-D3[57].

The Gibbs free energy change (G) can be calculated by: $G = E + E_{ZEP} - TS$, where E, $E_{ZPE}$ and S are the energy difference between the products and reactants from DFT calculations, the changes in zero-point energies and entropy at 25 °C, respectively. The reaction proceeded as follow:

$$\text{catalyst slab} + H_2O_2 \rightarrow \text{catalyst slab} *H_2O_2$$

$$\text{catalyst slab} *H_2O_2 \rightarrow \text{catalyst slab} *OH *OH$$

$$\text{catalyst slab} *OH *OH \rightarrow \text{catalyst slab} *OH + OH$$

$$\text{catalyst slab} *OH + OH \rightarrow \text{catalyst slab} + 2OH$$

The adsorption energy of $*H_2O_2$ ($\Delta E$) on the catalyst surface was calculated by: $\Delta E = E(*H_2O_2) - E(\text{slab}) - E(H_2O_2)$, in which $E(*H_2O_2)$, $E(\text{slab})$, and $E(H_2O_2)$ are the total energy of the catalyst slab with adsorbed $H_2O_2$, the catalyst slab and the energy of $H_2O_2$, respectively.

## Data availability

The data supporting the findings of the study are included in the main text and supplementary information files. Additional data can be obtained from the corresponding author upon request. The source data underlying Figs. 2–5 are provided with this paper as a Source Data file. Source data are provided with this paper.

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

## Acknowledgements

This work was funded by Australian Research Council (DP230102406 and FL230100178). W.T. acknowledges the partial support from the Australian Research Council Discovery Early Career Researcher Award (ARC-DECRA, DE220101074). The authors acknowledge funding and support from the Deutsche Forschungsgemeinschaft, under Germany´s Excellence Strategy — EXC 2089/1–390776260 e-conversion cluster, the Bavarian program Solar Energies Go Hybrid (SolTech) and the Center for NanoScience (CeNS). The authors acknowledge the scientific and technical assistance from Dr. Ashley Slattery at Adelaide Microscopy, the University of Adelaide. The authors also thank the soft X-ray spec-troscopy beamline and X-ray absorption spectroscopy beamline of Australian Synchrotron for supporting XANES and XAFS measurements. J.L. acknowledges Dr. Huanyu Jin and Dr. Roy Lehmann at the University of Adelaide for the help of online mass spectrometry analysis. Funding: This work was supported by Australian Research Council grants DP230102406, FL230100178 and DE220101074 and the Deutsche For-schungsgemeinschaft (DFG) under EXC 2089/1–390776260.

## Author contributions

W.T., H.Z., and S.W. supervised the project. J.L., W.T., and H.Z. conceived the original concept and initiated the project. J.L., K.H., and H.Z. con-ducted the experiments. J.L., K.H., W.T., T.H., X.D., H.S., H.Z., E.C., and S.W. performed data analyses. Y.W. performed the DFT computations. J.L., W.T., H.Z., and S.W. wrote the paper. All authors discussed the results and commented on the manuscript.

## Competing interests

The authors declare no competing interests.
