## [Peer Review File · Nature Communications]

Tandem microplastic degradation and hydrogen production by hierarchical carbon nitride-supported single-atom iron catalystsREVIEWER COMMENTS

Reviewer #1 (Remarks to the Author):

This study introduces a hierarchical CN-supported single-atom Fe (FeSA-hCN) catalyst and proposes a tandem strategy for upcycling microplastics and producing hydrogen. The FeSA-hCN demonstrates efficient degradation of plastic waste and showcases the potential for upgrading plastic conversion products to produce clean fuel, providing a cost-effective solution for controlling and utilizing plastic waste. The manuscript is well written and organized. Before accepted for publication, the author needs to consider the following questions and make certain revisions.

1. The " $\bullet\text{OH}$ " should be given its full name when first appearing in the manuscript (on line 72 instead of line 118)
2. It would be better if the name of atoms were labeled (Fig. 2h, Supplementary Fig. 8, Supplementary Fig. 11).
3. Please consider adding the results of " H_2O_2 only" to Fig. 3g, especially under $160\text{ }^\circ\text{C}$.
4. Reaction time (12 h?) should be specified in Fig. 3h, when the temperature is the horizontal axis title.
5. Line 149, "f, SEM images", the caption of Fig. 3f is not bolded.
6. Line 189, why is the last 2 runs performed under $160\text{ }^\circ\text{C}$ instead of $140\text{ }^\circ\text{C}$?
7. In Fig. 3d and Supplementary Fig. 32, the faster degradation rate of MPs, while the lower consumption of H_2O_2 (0-6 h vs 6-18 h). Please provide a reasonable explanation.
8. It is recommended to reflect the redox cycle of $\text{Fe}^{2+}/\text{Fe}^{3+}$ in Fig. 5i.
9. In order to more clearly reveal the photocatalytic hydrogen production mechanism in Fig. 6a, it is better to measure and analyze the energy band structure of the catalyst.
10. Is dicyandiamide and iron(III) acetylacetonate (line 379 and 386) uniformly dispersed on the silica template?

Reviewer #2 (Remarks to the Author):

This study tested the applicability of a nitride-supported single-atom iron catalysts in the degradation. They used a set of characterization to see the structures and performance for ultrahigh-molecular-weight- polyethylene, and revealed the good performance of this catalyst. The topic is interesting for me. However, authors did most of tests on catalyst. Instead, I may suggest author to add more tests and analyses on MPs to reveal more information or validation on its degradation mechanism. Therefore, authors need to largely improve the study.

1. In Fig. 2f, no characteristic peaks of Fe-C were observed in the FeSA-hCN material, while in Fig. 4c, the reaction-prior material in FeSA-hCN showed the diffraction peaks of Fe-C. There are reports in the literature that transition metals on carbonitride carriers tend to form M-N₄ and M-N₃C₁ (<https://doi.org/10.1038/s41929-0-22-00885-1>), so it is not rigorous that authors only built a model

of Fe-N4

- 2, Is the catalyst regenerable? If so, how can it be regenerated?
- 3, In the IR spectrum (Figure 3e), 1000-1300 cm^{-1} should correspond to the C-C and C-H bonds of UHMWPE. Why there is no obvious decrease during degradation?
- 4, After degradation, a variety of organic compounds have been produced, as listed in Table S6. Ensuring the safety of all degraded products is an essential requirement for a new catalyst. However, the analysis did not include an assessment of the ecological risks and potential ecotoxicological effects of these produced organics (like some organic organics). Therefore, I strongly recommend that the authors supplement their study with such an important test.
- 5, The selectivity of carboxylic acid has been a focal point in the study. Please elaborate on the significance of such selectivity. Additionally, a 64% selectivity does not appear to be particularly high to me.
- 6, 3. What are the dynamics of morphology, particle size, and structures of MPs during degradation? Understanding these factors is crucial for elucidating the degradation mechanism of MPs by the new material. Are there any differences in mechanisms compared to other catalysts reported in the literature?
- 7, “the hydrocarbon chains of UHMWPE were partially oxidized (C–H bond activation) by $\bullet\text{OH}$ ”...This can be easily verified by analyzing the surface oxygen-containing functional groups of degraded MPs.
- 8, “Tandem method” “Tandem process”...is unclear for me.
- 9, What are the particle sizes of the tested plastics or microplastics? Any pretreatment for the used microplastics? These information is missing.
- 10, It is unclear for me if authors did the control experiment (MPs degradation without catalyst under 140 and 160 $^{\circ}\text{C}$)? More details on QA/QC of degradation should be provided.

Reviewer #3 (Remarks to the Author):

This manuscript reported the Fe-N4 site in FeSA-hCN can effectively activate H_2O_2 to produce $\bullet\text{OH}$ for UHMWPE decomposition. In addition, the mixture of FeSA-hCN and plastic degradation products further achieves an ultrahigh hydrogen evolution of $42 \mu\text{mol h}^{-1}$ under illumination. This tandem process not only provides a scalable and economically feasible strategy to combat plastic pollution but also contributes to the hydrogen economy, with far-reaching implications for global sustainability initiatives.

However, I think it needs to be revised for publication, due to the lack of sufficient data and descriptions to support the conclusions presented in the manuscript.

1. The author should add the first derivative of the absorbing edge of different samples by XANES.
2. SEM images of adding different catalysts for hydrothermal plastics should be supplemented to explore the effects of different catalysts.
3. The EPR spectrum of superoxide radicals should be provided. Other free radicals may be generated during the reaction process.
4. The chemical formula of the reaction intermediate should be supplemented. In addition, the

Gibbs free energy of Fe-C₃N₄ in steps 4 and 5 is higher than that of C₃N₄, which leads to a decrease in activity.

5. During the reaction process, gas products will be generated, and the author should detect them.

Responses to reviewers' comments (Manuscript ID: NCOMMS-24-26541-T)

Title: Tandem microplastic degradation and hydrogen production by hierarchical carbon nitride-supported single-atom iron catalysts

Reviewer #1

This study introduces a hierarchical CN-supported single-atom Fe (FeSA-hCN) catalyst and proposes a tandem strategy for upcycling microplastics and producing hydrogen. The FeSA-hCN demonstrates efficient degradation of plastic waste and showcases the potential for upgrading plastic conversion products to produce clean fuel, providing a cost-effective solution for controlling and utilizing plastic waste. The manuscript is well written and organized. Before accepted for publication, the author needs to consider the following questions and make certain revisions.

Response to comments: We would like to thank the reviewer for the great comments and the positive recommendation for publication. We have considered all the comments carefully and tried our best to revise the manuscript accordingly. All the issues have been addressed point by point as shown below.

Comment 1-1. The " $\bullet\text{OH}$ " should be given its full name when first appearing in the manuscript (on line 72 instead of line 118)

Response and changes: Many thanks to you for pointing out this problem. Hydroxyl radical ($\bullet\text{OH}$) was presented when first appeared on page 3 and this full name was removed on page 5. The manuscript was further carefully proofread to eliminate typos and errors.

Page 3, the full name of hydroxyl radical was presented.

“The FeSA-hCN with Fe–N₄ sites effectively activates H₂O₂ to generate hydroxyl radical ($\bullet\text{OH}$) for breaking down ultrahigh molecular-weight polyethylene (UHMWPE).”

Comment 1-2. It would be better if the name of atoms were labeled (Fig. 2h, Supplementary Fig. 8, Supplementary Fig. 11).

Response: Many thanks to you for the great suggestion. The atom names were labelled in Fig. 2h, Supplementary Figs. 8 and 11 for clarification. All the figures were carefully checked.

Changes: *On page 4, atom names were added in Fig. 2h.*

Fig. 2 | h, The theoretical model of Fe-C₃N₄ for the representation of FeSA-hCN according to the Fe K-edge EXAFS analysis result.

Supplementary Materials, pages 10 and 14, atom names were added in Supplementary Figs. 8 and 12.

Supplementary Fig. 8 | Illustration of atomic structure of hydrogen-bonded polymeric melon-based carbon nitride.

Supplementary Fig. 12 | Simulated models for theoretical calculation. a, C_3N_4 and b, $Fe-C_3N_4$ for the representation of CN and FeSA-hCN.

Comment 1-3. Please consider add the results of "H₂O₂ only" to Fig. 3g, especially under 160 °C.

Response: Thank you for the great advice. The hydrothermal UHMWPE MPs degradation has been performed in the system of H₂O₂ only at 140 °C. We supply the result of the system with H₂O₂ only at 160 °C. A supplementary figure was added to give a clear presentation of the control experiments.

Changes: On page 5, the following content was added. (Last paragraph)

“For subsequent experiments, the FeSA-hCN with optimum 4.0 wt% Fe was used for MP degradation unless otherwise specified. Control experiments were conducted, including UHMWPE MP degradation in pure H₂O, FeSA-hCN only, and H₂O₂ only systems (Supplementary Fig. 21).”

Supplementary Materials, page 23, Supplementary Fig. 21 was added.

Supplementary Fig. 21 | UHMWPE MPs degradation performances in different control systems. Reaction conditions: [UHMWPE MPs] = 1 g L⁻¹, [catalyst] = 1 g L⁻¹ if used, [H₂O₂] = 100 mM if used, hydrothermal temperature = 140 or 160 °C, reaction time of 12 h and neutral pH.

Comment 1-4. Reaction time (12 h?) should be specified in Fig. 3h, when the temperature is the horizontal axis title.

Response: Thank you for pointing out the problem. The reaction time is indeed 12 h and we add such information in the inset in Fig. 3h.

Changes: Page 6, “12 h reaction” was added in inset of Fig. 3h.

Fig. 3 | h, TOC and IC concentrations.

Comment 1-5. Line 149, "f, SEM images", the caption of Fig. 3f is not bolded.

Response and changes: Thanks for pointing out the formatting issue and the caption was bolded. The manuscript was carefully proofread to eliminate any other typos and errors.

On page 6, the caption of Fig. 3f was bolded.

f, SEM images of fresh MPs and reaction residues after different reaction time (3, 6, 9 and 12 h)."

Comment 1-6. Line 189, why is the last 2 runs were performed under 160 °C instead of 140 °C?

Response: Thank you for raising this good question. This is because we want to illustrate that the recycled samples have good stability and reusability at both 140 and 160 °C, whether reused continuously or alternately. We add some explanations about the design of the cycling test experiments.

Changes: *On page 8, the following content was added.(Last paragraph)*

“To assess the recyclability and chemical stability, the catalyst was tested for 6 consecutive cycles (72 h) (Fig. 4a and Supplementary Fig. 32). Minor fluctuations were observed in reactivity after 4 cycles at 140 °C. The recovered catalysts after the 4th run were further tested for UHMWPE degradation at 160 °C, showing almost complete MP removal efficiency in the 5th and 6th cycling runs. This demonstrated the robust catalytic stability of FeSA-hCN, whether reused continuously or alternately at 140 and/or 160 °C.”

Comment 1-7. In Fig. 3d and Supplementary Fig. 32, the faster degradation rate of MPs, while the lower consumption of H₂O₂ (0-6 h vs 6-18 h). Please provide a reasonable explanation.

Response: Thank you for raising this incisive question. We believe that this phenomenon is related to the melting states of MPs. Our analysis indicates that MPs were gradually melted in 0-6 h. Although the consumption of H₂O₂ is fast in the first 6 h, the as-produced •OH has a limited effect on attacking the unmelted UHMWPE. After 6 h, UHMWPE MPs are mostly melted, making them easier to be oxidized by •OH. For the safety of the experiment, the measurement of H₂O₂ concentration was only conducted after the reactor was cooled down to room temperature. This may result in a certain delay in H₂O₂ detection. A more focus will be paid to investigating a better method to monitor the accurate change of H₂O₂ concentration in our future work. To observe the melting process of MPs, we supply SEM images

and photos taken on the UHMWPE collected after cooling down from different hydrothermal systems at 140 °C (Supplementary Figs. 24-27). Some analyses were added accordingly.

Changes: *On page 7, the following description was added. (Last paragraph)*

“Morphologies of UHMPWE were also monitored in different control systems of pure water, H₂O₂, and CN/H₂O₂ at 140 °C. UHMPWE aggregated with time in these systems (Supplementary Figs. 24-26) with smooth surfaces but some cracks occurred at 9 and 12 h in H₂O₂ and CN/H₂O₂, whereas bulk UHMPWE remained and cannot be effectively degraded as in FeSA-hCN/H₂O₂. The melting states of UHMPWE were then examined by observing the appearance changes of UHMWPE in pure water (Supplementary Fig. 27). UHMWPE MP powders aggregated into small fragments in 3 h, suggesting that they just started to melt. The fragments coalesced into a larger piece at 6 h, suggesting that more UHMWPE were melted. At 9 and 12 h, plastics formed a more uniform sheet with a smooth surface (Supplementary Fig. 24), indicating a further or full melting state under hydrothermal conditions. We speculate that the two-stage UHMWPE degradation kinetics in Fig. 3d is related to the melting states and the degradation is slow when UHMWPE is not adequately melted in 3 and 6 h, while the kinetics will be accelerated after 6 h when UHMWPE MPs are mostly melted.”

On page 10, the following description was added. (Last paragraph)

“While the consumption rate of H₂O₂ was fast in the first 3 h, the UHMWPE degradation rate was low in the first 6 h (Fig. 4c, d), which should be closely related to the melting states of MPs. As above-mentioned, UHMWPE was aggregated and was not adequately melted in 0-6 h, making it difficult for the as-produced •OH to attack. After 6 h, UHMWPE was mostly melted, making it easier to be oxidized by •OH. H₂O₂ was detected to be at 0.05 mM after 18 h at 140 °C, indicating that the amount of H₂O₂ was sufficient to support the production of •OH during the whole reaction time.”

Supplementary Materials, pages 26-29, Supplementary Figs. 24-27 were added.

Supplementary Fig. 24 | SEM images of UHMWPE MPs collected after reaction in a pure H₂O system at 140 °C. Reaction conditions: [UHMWPE MPs] = 1 g L⁻¹, hydrothermal temperature = 140 °C and neutral pH.

Supplementary Fig. 25 | SEM images of UHMWPE MPs collected after reaction in a control H₂O₂ only system at 140 °C. Reaction conditions: [UHMWPE MPs] = 1 g L⁻¹, [H₂O₂] = 100 mM, hydrothermal temperature = 140 °C and neutral pH.

Supplementary Fig. 26 | SEM images of UHMWPE MPs collected after reaction in a control CN/H₂O₂ system at 140 °C. Reaction conditions: [UHMWPE MPs] = 1 g L⁻¹, [CN] = 1 g L⁻¹, [H₂O₂] = 100 mM, hydrothermal temperature = 140 °C and neutral pH.

Supplementary Fig. 27 | Photos of UHMWPE MPs after reaction in a pure H₂O system at 140 °C. Reaction conditions: [UHMWPE MPs] = 1 g L⁻¹, hydrothermal temperature = 140 °C and neutral pH.

Comment 1-8. It is recommended to reflect the redox cycle of Fe²⁺/Fe³⁺ in Fig. 5i.

Response and changes: Thank you for the great suggestion. The illustration of the Fe²⁺/Fe³⁺ redox cycle was added in Fig. 5i. (page 11)

Fig. 5 | i, UHMWPE degradation mechanism.

Comment 1-9. In order to more clearly reveal the photocatalytic hydrogen production mechanism in Fig. 6a, it is better to measure and analyze the energy band structure of the catalyst.

Response: Thank you for the valuable suggestion. As FeSA-hCN after hydrothermal degradation of UHMWPE was directly used for hydrogen production, we supply its light absorption property by UV/Vis diffuse reflectance spectra. The energy band structure was also analyzed by Tauc plots obtained from the Kubelka–Munk function, while the valence state position was determined by the valence band X-ray photoelectron spectroscopy.

Changes: On page 14, the band structure of FeSA-hCN was updated in Fig. 6a. (Page 14)

Fig. 6 | Photocatalytic hydrogen production using reaction product mixture. a, Illustration of photocatalytic hydrogen production mechanism using the reaction product mixture after hydrothermal degradation of UHMWPE MPs and the band structure of FeSA-hCN, determined based on Supplementary Fig. 53.

On page 20, the following content was added. (First paragraph)

“Electronic band structure study

UV-vis diffuse reflectance spectra were measured using a Cary 100 UV-vis spectrophotometer (Agilent, US). The bandgap of FeSA-hCN catalyst after hydrothermal UHMWPE degradation was estimated by the Tauc plots, obtained from the Kubelka-Munk function. The valence state position was determined by the valence band X-ray photoelectron spectroscopy (XPS, Thermo Scientific K-Alpha).”

Supplementary Materials, page 55, Supplementary Fig. 53 with some illustrations were supplied.

Supplementary Fig. 53 | Electronic structure analysis of catalysts after hydrothermal UHMWPE degradation. a, UV/Vis DRS. b, Tauc plot. c, valence band X-ray photoelectron spectrometer (XPS). d, Electronic band structure.

The light absorption of the catalyst (collected after the hydrothermal UHMWPE degradation) using UV-vis diffusion reflection spectra (DRS). The bandgap energy (about 2.34 eV) was determined by the transformational Tauc plots obtained from the Kubelka-Munk function. The valence state was determined using the valence band X-ray photoelectron spectrometer (XPS). The electronic band structure of the catalyst is shown in Supplementary Fig. 53d and Fig. 6a.

Comment 1-10. Is dicyandiamide and iron(III) acetylacetonate (line 379 and 386) uniformly dispersed on the silica template?

Response: Thank you for pointing out the unclear description. The mixed dicyandiamide and iron (III) acetylacetonate ($\text{Fe}(\text{acac})_3$) were tiled just on the surface of the silica templates. We also modified the description in the **Materials and Methods** section to make the clear information.

Changes: On page 17, the content was revised as shown below.

“The as-obtained solution was dropped into vials (10 mL) for ultrasonication and evaporation at 110 °C for 24 h. The silica microspheres were self-assembled on the wall of the vials, forming silica strips.” (First paragraph)

“The as-obtained solids were collected and ground for further use. Then, the well-mixed solids were uniformly tiled on the surface of 1.0 g silica template strips, calcined at 520 °C for 2 h with a ramp of 2 °C·min⁻¹ and further to 550 °C for 2 h with a ramp of 4 °C min⁻¹ under N₂ atmosphere.” (Fourth paragraph)

Reviewer #2

This study tested the applicability of a nitride-supported single-atom iron catalysts in the degradation. They used a set of characterization to see the structures and performance for ultrahigh-molecular-weight-polyethylene, and revealed the good performance of this catalyst. The topic is interesting for me. However, authors did most of tests on catalyst. Instead, I may suggest author to add more tests and analyses on MPs to reveal more information or validation on its degradation mechanism. Therefore, authors need to largely improve the study.

Response to comments: We thank you for the positive comments that helped us improve the quality of this work and gave us a lot of inspiration. We have carefully revised the manuscript and fully addressed the comments. We supply the characterizations of UHMWPE MPs, including FTIR spectra, SEM images, photos, and size distributions before and after Fenton-like catalytic degradation, as shown in the below response to the raised questions.

Comment 2-1. In Fig. 2f, no characteristic peaks of Fe-C were observed in the FeSA-hCN material, while in Fig. 4c, the reaction-prior material in FeSA-hCN showed the diffraction peaks of Fe-C. There are reports in the literature that transition metals on carbonitride carriers tend to form $M-N_4$ and $M-N_3C_1$ (<https://doi.org/10.1038/s41929-0-22-00885-1>), so it is not rigorous that authors only built a model of $Fe-N_4$.

Response: Thank you for the great suggestions and valuable reference information. In EXAFS spectra (Figs. 2f and 4c), FeSA-hCN displayed a dominant peak relating to Fe-N(C) coordination. We further adopted quantitative EXAFS curve fitting analysis, which indicated that the first shell coordination configuration was determined as $Fe-N_4$ structure. To verify this point and exclude other Fe-N(C) coordinations (e.g., FeN_2C_2), we supplied theoretical simulations and established several Fe-N(C) coordination configurations. After optimization, the $Fe-N_4$ structure by coordinating the Fe atom with three triangular edge N (sp^2-N_b) and one amino group/bridging N (sp^3-N_a) shows more negative formation energy than others, indicating that it is thermodynamically more favorable to exist (Fig. 2h, Supplementary Fig. 12). This is because, in a CN structure, the N atoms are recognized as the electron-abundant sites to provide rich electron lone pairs to incorporate with single metal ions. The theoretical $Fe-N_4$ model is consistent with our experimental results.

Changes: *On page 5, the content was modified as shown below. (Second paragraph)*

“Fourier transform (FT) extended X-ray absorption fine structure spectroscopy (EXAFS) spectra displayed a dominant peak near 1.6 Å, ascribed to the **Fe-N(C) first-shell coordination** (Fig. 2f)³². No Fe-Fe interactions (2.2 Å) were observed, confirming the atomic dispersion feature of Fe atoms in FeSA-hCN. The quantitative EXAFS fitting curves (Fig. 2f and Supplementary Fig. 11) and the corresponding fitting results (Supplementary Table 5) revealed a $Fe-N_4$ coordination with an average bond distance of 2.1 Å. The wavelet transform (WT) contour plot further demonstrates the first-shell Fe-N pattern (Fig. 2g). **Considering the possible metal-N(C) first-shell coordination in carbon nitride carriers^{32,33}, we established and optimized several theoretical structure models by density functional theory (DFT) calculations (Supplementary Table 6). Three types of N (N_a , N_b , N_c , Supplementary Fig. 12) exist. The $Fe-N_4$ structure by coordinating the Fe atom with three triangular edge N (sp^2-N_b) and one amino**

group/bridging N (sp^3-N_a) was thermodynamically favorable to form (Fig. 2h, Supplementary Fig. 12), which is consistent with the experimental results. DFT simulation also demonstrated that the Fe–N₄ sites could modulate the electronic structure of carbon nitride through charge redistribution (Supplementary Fig. 13), affecting its catalytic performance.”

References

32. Pan, Y. *et al.* Regulating the coordination structure of single-atom Fe-N_xC_y catalytic sites for benzene oxidation. *Nat. Commun.* **10**, 4290 (2019).
33. Zhou, Y. *et al.* Peripheral-nitrogen effects on the Ru₁ centre for highly efficient propane dehydrogenation. *Nat. Catal.* **5**, 1145–1156 (2022).

Supplementary Materials, page 14, N_a, N_b, N_c, were labeled in Supplementary Fig. 12.

Supplementary Fig. 12 | Simulated models for theoretical calculation. a, C₃N₄ and b, Fe-C₃N₄ for the representation of CN and FeSA-hCN.

Supplementary Materials, pages 61-62, Supplementary Table 6 and some illustrations were added.

Supplementary Table 6 | Theoretical models established for FeSA-hCN.

Model	Before optimization	After optimization	Bond length	Formation energy
1			$d_1=2.098 \text{ \AA}$ $d_2=2.069 \text{ \AA}$ $d_3=2.188 \text{ \AA}$ $d_4=2.055 \text{ \AA}$ $d_{ave}=2.103 \text{ \AA}$	-3.7796 eV
2			$d_1=2.079 \text{ \AA}$ $d_2=2.069 \text{ \AA}$ $d_3=2.361 \text{ \AA}$ $d_4=2.109 \text{ \AA}$ $d_{ave}=2.155 \text{ \AA}$	-3.7794 eV
3			$d_1=2.096 \text{ \AA}$ $d_2=2.083 \text{ \AA}$ $d_3=2.353 \text{ \AA}$ $d_4=2.113 \text{ \AA}$ $d_{ave}=2.161 \text{ \AA}$	-3.4331 eV
4			$d_1=2.041 \text{ \AA}$ $d_2=2.048 \text{ \AA}$ $d_3=1.928 \text{ \AA}$ $d_4=1.921 \text{ \AA}$ $d_{ave}=1.985 \text{ \AA}$	-2.8838 eV
Notes: 				

The thermal condensation of DCD at 550 °C promoted the formation of a hydrogen-bonded polymeric melon-based carbon nitride structure with abundant NH/NH₂ groups (Supplementary Figs. 7 and 8). This provides an ideal framework for stabilizing single-atom metals between the melon chain by bonding with surrounding C and N atoms, especially electron-abundant N sites with rich electron lone pairs. Considering the possible Fe–N(C) coordination configurations in a melon-structured carbon nitride

framework, we established four theoretical models. After the optimization using DFT calculation, it was found that Fe atoms tend to coordinate with the surrounding N atoms to form a Fe–N₄ structure. As illustrated in Supplementary Fig. 12a, there exist three types of N (N_a, N_b, N_c, Supplementary Fig. 12) in such a type of carbon nitride structure, i.e., amino groups/bridging N (sp³-N_a), triangular edge N (sp²-N_b), and central tertiary N (N_c). The DFT results indicated that the optimized Fe–N₄ ‘model 1’ (by coordinating with three triangular edge N (sp²-N_b) and one amino group/bridging N (sp³-N_a)) had negatively lower formation energy than the others, and therefore, is thermodynamically favorable to exist.

Comment 2-2. Is the catalyst regenerable? If so, how can it be regenerated?

Response: Thank you for raising the questions. As shown in Fig. 4a, the catalyst is regenerable and reusable with high catalytic activity over 6 cycling runs (72 h at 140 and 160 °C). The FeSA-hCN catalyst is regenerated by filtration of the reaction solution after cooling down to room temperature, and washed alternately with water and ethanol.

Changes: On page 8, the following content was modified.(Last paragraph)

“To assess the recyclability and chemical stability, the catalyst was tested for 6 consecutive cycles (72 h) (Fig. 4a and Supplementary Fig. 32). Minor fluctuations were observed in reactivity after 4 cycles at 140 °C. The recovered catalysts after the 4th run were further tested for UHMWPE degradation at 160 °C, showing almost complete MP removal efficiency in the 5th and 6th cycling runs. This demonstrated the robust catalytic stability of FeSA-hCN, whether reused continuously or alternately at 140 and/or 160 °C.”

Pages 18-19, the methods for cycling tests, including the regeneration for FeSA-hCN during cycling tests, were added in Materials and Methods.

“Cycling UHMWPE degradation tests

The dried reaction residues on the CA membrane were dispersed in a 70 mL beaker with 50 mL ultrapure water. After sitting for 24 h, most unreacted UHMWPE tended to float over water. These UHMWPE samples were carefully removed using tweezers. The remaining FeSA-hCN catalysts were collected by filtration, washed alternately with water and ethanol, and dried at 60 °C for the next cycling run.”

Comment 2-3. In the IR spectrum (Figure 3e), 1000-1300 cm⁻¹ should correspond to the C-C and C-H bonds of UHMWPE. Why there is no obvious decrease during degradation?

Response: Thank you for raising this incisive question. Fig. 3e delivers the FTIR spectrum of FeSA-hCN and the overlapped spectra of reacted UHMWPE/FeSA-hCN. We further supplied and analyzed the FTIR spectra of pure and reacted UHMWPE in Supplementary Fig. 23. Comparing Fig. 3e and Supplementary Fig. 23, we can deduce that the peaks between 1000-1800 cm⁻¹ (this includes 1000-1300 cm⁻¹) in Fig. 3e mainly correspond to C–N heterocycles in FeSA-hCN. The peaks at around 718, 2847, and 2915 cm⁻¹ correspond to the vibrations of rocking deformation C–H, symmetric –CH₂– stretching, and asymmetric –CH₂– stretching groups of UHMWPE, respectively. With the prolonged reaction time, the intensity of these peaks gradually declined.

Changes: On page 6, Fig. 3e caption was modified as shown below.

Fig. 3 e, FT-IR spectra of **FeSA-hCN** and **FeSA-hCN/UHMWPE** reaction residues after 140 °C hydrothermal reaction at different resistance time. Marked bands in 1000 to 1800 cm^{-1} correspond to C-N heterocycles in carbon nitride (as compared to the pure UHMWPE in Supplementary Fig. 23).

Page 7 (Second paragraph)

“FTIR spectra in Fig. 3e and **Supplementary Fig. 23** showed three main peaks at around 718, 2847, and 2915 cm^{-1} , corresponding to the vibrations of rocking deformation C-H, symmetric $-\text{CH}_2-$ stretching, and asymmetric $-\text{CH}_2-$ stretching groups of UHMWPE, respectively.”

On page 19, the following FTIR measurement methods were added.

“FTIR analysis of reaction residues

The dried reaction residues on the CA membrane were carefully transferred to an agate mortar, followed by grinding. The well-mixed solids were measured by a Nicolet 6700 FTIR Spectrometer (Thermofisher).

FTIR analysis of UHMWPE after reaction

After reactions with different time, the UHMWPE was carefully removed from the Teflon reactor using tweezers. The UHMWPE was collected using a 500 mL beaker and washed with 300 mL water to remove catalysts on the UHMWPE surface. After magnetically stirring at 600 rpm for 2h, the UHMWPE was collected by vacuum filtration. After drying, the UHMWPE was analyzed using the Nicolet 6700 FTIR Spectrometer (Thermofisher).”

Supplementary Materials, page 25, Supplementary Fig. 23 was added.

Notes:

- a. In-plane CH_2 bending vibration
- b. In-plane CH_3 asymmetric bending
- c. CH_2 group adjacent to the carbonyl group in ketones
- d. CH_3 asymmetric deformation
- e. CH_2 out of plane vibration
- f. C-O stretching
- g. C-O-C
- h. O-H out of plane bending vibration in COOH

Supplementary Fig. 23 | FTIR analysis of UHMWPE MPs separated and collected after reactions at different time in FeSA-hCN/ H_2O_2 .

Comment 2-4. After degradation, a variety of organic compounds have been produced, as listed in Table S6. Ensuring the safety of all degraded products is an essential requirement for a new catalyst. However,

the analysis did not include an assessment of the ecological risks and potential ecotoxicological effects of these produced organics (like some organic organics). Therefore, I strongly recommend that the authors supplement their study with such an important test.

Response: We thank you the great recommendation for the study of the ecological risks and potential ecotoxicological effects of our reaction solution. The ecotoxicities (including acute and chronic toxicities) of the organic products were further assessed and evaluated by the Ecological Structure Activity Relationships 19 (ECOSAR) Software.

Changes: *On page 13, the following content was added. (First paragraph)*

“The ecotoxicities (including acute and chronic toxicities to fish, daphnids, and green algae) of the possible organic products were evaluated using the Ecological Structure Activity Relationships 19 (ECOSAR) Software (Supplementary Figs. 51 and 52). The majority of reaction intermediates and products showed low acute and chronic toxicities to fish, daphnids, and green algae. Our FeSA-hCN/H₂O₂ Fenton-like system excels in MP degradation efficiency, catalyst stability, versatility, and mild pH operation conditions, setting a new benchmark beyond the currently documented literature (Supplementary Table 1).”

On page 20, the following analysis was added. (Third paragraph)

“An Agilent Chemstation software was used to undertake the data interpretation with the NIST14 mass spectral library as the database. The acute and chronic toxicities of reaction intermediates and products were evaluated by the Ecological Structure Activity Relationships 19 (ECOSAR) Software, developed by the United States Environmental Protection Agency.”

Supplementary Materials, pages 53-54, Supplementary Figs. 51, 52 were added.

Supplementary Fig. 51 | Acute ecotoxicities of possible organic products to fish, daphnids, and green algae via Ecological Structure Activity Relationships 19 (ECOSAR) Software. a, fish. b, daphnids. c, green algae (Organic chemicals label number as shown in Supplementary Table 8).

Supplementary Fig. 52 | Chronic ecotoxicities of possible organic products to fish, daphnids, and green algae via Ecological Structure Activity Relationships 19 (ECOSAR) Software. a, fish. b, daphnids. c, green algae (Organic chemicals label number as shown in Supplementary Table 8).

Comment 2-5. The selectivity of carboxylic acid has been a focal point in the study. Please elaborate on the significance of such selectivity. Additionally, a 64% selectivity does not appear to be particularly high to me.

Response: Thank you for raising this concern. As mentioned in the introduction, in previously reported Fenton/Fenton-like systems, the plastics degradation mechanism and degradation intermediates/product species were rarely investigated. As summarized in Supplementary Table 1, product selectivity has never been reported. In this study, we fill this gap by analyzing the reaction intermediates and products. We also offer new insights into the potential use of the main intermediates/products, carboxylic acids, in

solar fuel production. We agree that product selectivity at 64% is not very high at the current stage. Our future studies will focus on designing new catalysts and systems to achieve a higher selectivity in microplastic oxidation. For instance, this may be achieved by pretreating the plastics to introduce functional groups, allowing the radicals to attack at certain sites.

Changes: *Some revisions are made on page 13.*

“Future studies will be made toward further optimizing the process’s efficiency and selectivity, exploring its practical viability in large-scale applications.” (First paragraph)

“Following the examination of plastic degradation products, we now assess their practical value in solar fuel production. The predominant presence of carboxylic acids (product selectivity of 64%) in the UHMWPE degradation products has led to their consideration as potential sacrificial agents for sustainable photocatalytic hydrogen production (Fig. 6a).” (Second paragraph)

Comment 2-6. What are the dynamics of morphology, particle size, and structures of MPs during degradation? Understanding these factors is crucial for elucidating the degradation mechanism of MPs by the new material. Are there any differences in mechanisms compared to other catalysts reported in the literature?

Response: Thank you for the great questions. The SEM images in Fig. 3f show information on the morphology, size, and structure changes of UHMWPE MPs during degradation by FeSA-hCN/H₂O₂. For comparison, morphology changes in other systems (H₂O₂ only, and CN/H₂O₂) were also investigated by SEM images (Supplementary Figs. 24-26). Besides, the appearance of UHMWPE MP changes (Supplementary Fig. 27) after reactions at different time in a pure H₂O system was observed. The mechanism differences between our system with other catalysts reported in the literature were discussed.

Changes: *On page 7, the following analyses of UHMPWE morphology, size, and structure changes were added.*

“Morphological changes of UHMWPE degradation residues in FeSA-hCN/H₂O₂ system at 140 °C were shown in Fig. 3f: (i) within the initial 3 h, UHMWPE MPs aggregated into larger clumps; (ii) by the 6-h mark, UHMWPE pieces were cracked and fragmented; (iii) after 9-h reaction, the fragmentation degree was enhanced with newly formed cracks; (iv) by 12 h, most UHMWPE MPs were degraded and only smaller particle residues were collected.

Morphologies of UHMPWE were also monitored in different control systems of pure water, H₂O₂, and CN/H₂O₂ at 140 °C. UHMPWE aggregated with time in these systems (Supplementary Figs. 24-26) with smooth surfaces but some cracks occurred at 9 and 12 h in H₂O₂ and CN/H₂O₂, whereas bulk UHMPWE remained and cannot be effectively degraded as in FeSA-hCN/H₂O₂. The melting states of UHMPWE were then examined by observing the appearance changes of UHMWPE in pure water (Supplementary Fig. 27). UHMWPE MP powders aggregated into small fragments in 3 h, suggesting that they just started to melt. The fragments coalesced into a larger piece at 6 h, suggesting that more UHMWPE were melted.

At 9 and 12 h, plastics formed a more uniform sheet with a smooth surface (Supplementary Fig. 24), indicating a further or full melting state under hydrothermal conditions. We speculate that the two-stage UHMWPE degradation kinetics in Fig. 3d is related to the melting states and the degradation is slow when UHMWPE is not adequately melted in 3 and 6 h, while the kinetics will be accelerated after 6 h when UHMWPE MPs are mostly melted.”

Supplementary Figs. 24-27 were supplied.

Supplementary Fig. 24 | SEM images of UHMWPE MPs collected after reaction in a pure H₂O system at 140 °C. Reaction conditions: [UHMWPE MPs] = 1 g L⁻¹, hydrothermal temperature = 140 °C and neutral pH.

Supplementary Fig. 25 | SEM images of UHMWPE MPs collected after reaction in a control H₂O₂ only system at 140 °C. Reaction conditions: [UHMWPE MPs] = 1 g L⁻¹, [H₂O₂] = 100 mM, hydrothermal temperature = 140 °C and neutral pH.

Supplementary Fig. 26 | SEM images of UHMWPE MPs collected after reaction in a control CN/H₂O₂ system at 140 °C. Reaction conditions: [UHMWPE MPs] = 1 g L⁻¹, [CN] = 1 g L⁻¹, [H₂O₂] = 100 mM, hydrothermal temperature = 140 °C and neutral pH.

The mechanism comparison between our system with counterpart systems in literature was discussed on page 8. (Second paragraph)

“Different from previous homogenous Fenton processes that reported a high mineralization conversion of MPs into CO₂ and H₂O^{15,36,37}, total organic carbon (TOC) tests in FeSA-hCN/H₂O₂ system show that the TOC concentration in the reaction mixture slightly increased to 66.6 mg L⁻¹ in the first 9 h and surged to 652.8 mg L⁻¹ with the prolonged reaction time to 18 h at 140 °C (Fig. 3h and Supplementary Fig. 29). A higher TOC concentration (700.5 mg L⁻¹) was observed after a 12-h reaction at 160 °C. This growth trend was consistent with the weight loss results of MPs, suggesting that UHMWPE MPs were converted into liquid organic carbon species. We also tried to detect gas products. Only trace CO₂ and H₂ were detected after the reaction, and their amounts were far lower than the organic chemicals in the solution (Supplementary Fig. 30). Signals of acetylene (C₂H₂) and methane (CH₄) were also found; however, their intensities were below the analytical limit. TOC and gas product analysis further verified that the UHMWPE MPs were mostly converted into organic chemicals (Supplementary Fig. 31).”

Page 10 (Second paragraph)

“To date, different Fenton-like systems have been reported including persulfate-based AOP¹⁴, thermal-assisted Fenton reaction^{15,37}, photo-Fenton reaction⁴⁰, and electro-Fenton-like reaction¹⁶ to degrade MPs⁸. The degradation mechanisms vary based on different configurations of catalysts, oxidants, irradiation, and electrochemistry, producing different ROS (e.g., sulfate radical, superoxide radical (O₂⁻), •OH) or involving photo-induced holes or electron transfer⁸.”

Supplementary Materials, pages 32-33, Supplementary Figs. 30 and 31 were added.

Supplementary Fig. 30 | Detection of gas produced after the hydrothermal UHMWPE MPs degradation using the 8890 GC-TCD detector. Reaction conditions: reaction time = 18 h, [UHMWPE MPs] = 1 g L⁻¹, [catalysts] = 1 g L⁻¹ if used, [H₂O₂] = 100 mM if used, hydrothermal temperature = 140 or 160 °C and neutral pH.

After 18 h hydrothermal degradation of UHMWPE, the amount of CO₂ in the system was determined as 29.6 mg L⁻¹, and H₂ was detected as 0.4 mg L⁻¹.

Supplementary Fig. 31 | Detection of gas produced during hydrothermal UHMWPE MPs degradation using the 8890 GC-FID detector. Reaction conditions: reaction time = 18 h, [UHMWPE MPs] = 1 g L⁻¹ if present, [catalysts] = 1 g L⁻¹ if used, [H₂O₂] = 100 mM if used, hydrothermal temperature = 140 or 160 °C and neutral pH.

Only small signals of acetylene (C_2H_2) and methane (CH_4) were detected, which are below the analytical limits.

Comment 2-7. “the hydrocarbon chains of UHMWPE were partially oxidized (C–H bond activation) by •OH”...This can be easily verified by analyzing the surface oxygen-containing functional groups of degraded MPs.

Response: Thank you for the good suggestion. To analyze the surface functional groups, we carefully separated and collected UHMWPE MPs after reaction for FTIR tests. Peaks of C=O and C–O appeared and increased with prolonged time at 3, 6, and 9 h, suggesting the oxidation of C–H by •OH.

Changes: Page 12, the following changes were made. (Last paragraph)

“Early reaction intermediates (retention time around 4 minutes) were identified as signals corresponding to the formation of ketone functional groups (C=O). Combined with the EPR and quenching experiments, it could be concluded that the hydrocarbon chains of UHMWPE were partially oxidized (C–H bond activation) by •OH in the induction period (Fig. 5i)⁴⁴. This was verified by the FTIR analysis of

UHMWPE collected after the reaction. C=O and C–O groups appeared and gradually increased with the prolonged reaction time (Supplementary Fig. 23).”

Page 19, the following content was added.

“FTIR analysis of UHMWPE after reaction

After reactions with different time, the UHMWPE was carefully removed from the Teflon reactor using tweezers. The UHMWPE was collected using a 500 mL beaker and washed with 300 mL water to remove catalysts on the UHMWPE surface. After magnetically stirring at 600 rpm for 2h, the UHMWPE was collected by vacuum filtration. After drying, the UHMWPE was analyzed using the Nicolet 6700 FTIR Spectrometer (Thermofisher).”

Comment 2-8. “Tandem method” “Tandem process”...is unclear for me.

Response and changes: Thank you for pointing out the unclear statement. The tandem method and tandem process was revised as: tandem catalytic microplastic degradation-hydrogen evolution reaction (MPD-HER) process. Corresponding changes were made.

Pages 1-3, the following changes were made.

“In tackling microplastic pollution and advancing green hydrogen production, this study reveals a tandem catalytic microplastic degradation-hydrogen evolution reaction (MPD-HER) process using hierarchical porous carbon nitride-supported single-atom iron catalysts (FeSA-hCN).” (Abstract, first sentence)

“This tandem MPD-HER process not only provides a scalable and economically feasible strategy to combat plastic pollution but also contributes to the hydrogen economy, with far-reaching implications for global sustainability initiatives. Future research will focus on overcoming practical application barriers, broadening the range of degradable plastics, and further optimizing the process's efficiency and selectivity.” (Abstract, last two sentences)

Fig. 1 | Generation of microplastic wastes, potential harms, and the proposed tandem microplastic degradation-hydrogen evolution reaction (MPD-HER) process strategy.

“This study introduces a tandem catalytic MPD-HER process (Fig. 1, bottom) by integrating MPs degradation and photocatalytic H₂ production using a hierarchical CN-supported single-atom Fe catalyst (FeSA-hCN).” (page 3, third paragraph)

Comment 2-9. What are the particle sizes of the tested plastics or microplastics? Any pretreatment for the used microplastics? These information is missing.

Response: Thank you for pointing out this problem. We supplied the size information of the MPs. Plastic pellets, including PET, HDPE, LDPE, PS and PP, were smashed by a grinder to get MPs. For real-life

plastic products, plastic bags, food storage, drinking bottles, wet wipes bottles, and medical ziplock bags were cut into debris (<5 mm). No other pretreatment of MPs is involved.

Changes: *The following changes were made on page 10. (First paragraph)*

“We then investigated the degradation performance of FeSA-hCN on different polymer types (Fig. 4f and size information is presented in Supplementary Fig. 38), namely, polyethylene terephthalate (PET), high-density polyethylene (HDPE), polyvinyl chloride (PVC), low-density polyethylene (LDPE), polypropylene (PP) and polystyrene (PS).”

Page 16 (Second paragraph)

“Pretreatment of plastics

PET, HDPE, LDPE, and PP pellets were smashed by a grinder (IKA A 11 basic analytical mill, grinding for 10 s every 10 min). PS microplastics were prepared via ball-milling of PS pellets using a planetary mill (FRITSCH PULVERISETTE 7 with zirconia balls and vials) at a speed of 300 rpm overnight, operating for 2 min in every 7 min. The obtained particles were filtered by a 100 µm stainless steel sieve. Real-life plastic bags, food storage, drinking bottles, wet wipes bottles, and medical ziplock bags were cut into debris (< 5 mm). After rinsing with ethanol and water, the plastics were dried in air for 3 days. The particle sizes of commercial UHMWPE, PET, HDPE, PVC, LDPE, PP, and PS powders were analyzed using the Mastersizer 2000 - Malvern.”

Supplementary Materials, page 40, Supplementary Fig. 38 was added.

	UHMW PE	PET	HDPE	PVC	LDPE	PP	PS
D(0.1) μm	21.3	74.4	77.2	133.1	34.1	21.4	35.5
D(0.5) μm	39.6	104.0	108.0	190.2	89.3	60.2	112.3
D(0.9) μm	69.7	145.0	150.1	271.2	176.1	120.4	243.3

Notes:

D(0.1) μm: 10% particles in the tested powders are smaller than this size.

D(0.5) μm: 50% particles in the tested powders are smaller than this size.

D(0.9) μm: 90% particles in the tested powders are smaller than this size.

Supplementary Fig. 38 | Size distribution analysis of different microplastics.

Comment 2-10. It is unclear for me if authors did the control experiment (MPs degradation without catalyst under 140 and 160 °C)? More details on QA/QC of degradation should be provided.

Response: Thank you for pointing out the unclear description. We agreed that robust QA/QC is imperative to ensure the quality and consistency of experimental results. Control experiments have been performed. To give a clear presentation, we added a figure to compare results in different control systems, including pure H₂O, FeSA-hCN only, and H₂O₂ only at both 140 and 160 °C.

Changes: On page 6, the following changes were made.

For subsequent experiments, the FeSA-hCN with optimum 4.0 wt% Fe was used for MP degradation unless otherwise specified. Control experiments were conducted, including pure H₂O, FeSA-hCN only, and H₂O₂ only (Supplementary Fig. 21).

Supplementary Materials, page 23, Supplementary Fig. 21 was added.

Supplementary Fig. 21 | UHMWPE MPs degradation performances in different control systems. Reaction conditions: [UHMWPE MPs] = 1 g L⁻¹, [catalyst] = 1 g L⁻¹ if used, [H₂O₂] = 100 mM if used, hydrothermal temperature = 140 or 160 °C, reaction time of 12 h and neutral pH.

Reviewer #3

This manuscript reported the Fe-N₄ site in FeSA-hCN can effectively activate H₂O₂ to produce •OH for UHMWPE decomposition. In addition, the mixture of FeSA-hCN and plastic degradation products further achieves an ultrahigh hydrogen evolution of 42 μmol h⁻¹ under illumination. This tandem process not only provides a scalable and economically feasible strategy to combat plastic pollution but also contributes to the hydrogen economy, with far-reaching implications for global sustainability initiatives. However, I think it needs to be revised for publication, due to the lack of sufficient data and descriptions to support the conclusions presented in the manuscript.

Response to comments: We thank the reviewer for the encouraging appraisal and great comments, which helped us improve the quality of this work. All raised comments have been addressed point by point as shown below.

Comment 3-1. The author should add the first derivative of the absorbing edge of different samples by XANES.

Response: Thank you for the great suggestion. We have provided the first derivative of the absorbing edge of fresh and used FeSA-hCN. We admit that comparing it with other reference samples is important to help estimate the oxidation state of Fe. The first derivative curves of Fe K-edge XANES of reference Fe foil, FeO, and Fe₂O₃ were added. The absorption edge energy (E₀) - the value of the first maxima in the first-derivative spectra is typically used to reflect the oxidation state. The values of E₀ of different samples were compared to estimate the oxidation state of Fe in FeSA-hCN.

Changes: *The following changes were made on page 5.(First paragraph)*

“The chemical states and local coordination configuration of single-atom Fe in FeSA-hCN were further examined. The L-edge XANES spectra (Supplementary Fig. 9) evidenced that Fe mainly existed as Fe²⁺ (Supplementary Table 4)³⁰. **The Fe K-edge XANES and corresponding first derivatives further confirmed that the oxidation state of Fe in FeSA-hCN is close to 2 (Fig. 2e and Supplementary Fig. 10).**”

Supplementary Materials, page 12, Supplementary Fig. 10 was added.

Supplementary Fig. 10 | First derivatives of Fe K-edge XANES and oxidation states estimation.

Comment 3-2. SEM images of adding different catalysts for hydrothermal plastics should be supplemented to explore the effects of different catalysts.

Response: Thank you for the inspiring suggestions. SEM images of different systems, including pure water, H₂O₂ only, and CN/H₂O₂, were added to explore the effects of different systems on hydrothermal plastic degradation (Supplementary Figs. 24-26). Appearance changes of UHMWPE in pure water were also observed by taking photos of the separated UHMWPE (Supplementary Fig. 27).

Changes: *The following changes were made on page 7. (Last paragraph)*

“Morphologies of UHMPWE were also monitored in different control systems of pure water, H₂O₂, and CN/H₂O₂ at 140 °C. UHMPWE aggregated with time in these systems (Supplementary Figs. 24-26) with smooth surfaces but some cracks occurred at 9 and 12 h in H₂O₂ and CN/H₂O₂, whereas bulk UHMPWE remained and cannot be effectively degraded as FeSA-hCN/H₂O₂. The melting states of UHMPWE were then examined by observing the appearance changes of UHMWPE in pure water (Supplementary Fig. 27). UHMWPE MP powders aggregated into small fragments in 3 h, suggesting that they just started to melt. The fragments coalesced into a larger piece at 6 h, suggesting that more UHMWPE was melted. At 9 and 12 h, plastics formed a more uniform sheet with a smooth surface (Supplementary Fig. 24), indicating a further or full melting state under hydrothermal conditions. We speculate that the two-stage UHMWPE degradation kinetics in Fig. 3d is related to the melting states and the degradation is slow when UHMWPE is not adequately melted in 3 and 6 h, while the kinetics will be accelerated after 6 h when UHMWPE MPs are mostly melted.”

Supplementary Materials, pages 26-29, Supplementary Figs. 24-27 were added.

Supplementary Fig. 24 | SEM images of UHMWPE MPs collected after reaction in a pure H₂O system at 140 °C. Reaction conditions: [UHMWPE MPs] = 1 g L⁻¹, hydrothermal temperature = 140 °C and neutral pH.

Supplementary Fig. 25 | SEM images of UHMWPE MPs collected after reaction in a control H₂O₂-only system at 140 °C. Reaction conditions: [UHMWPE MPs] = 1 g L⁻¹, [H₂O₂] = 100 mM, hydrothermal temperature = 140 °C and neutral pH.

Supplementary Fig. 26 | SEM images of UHMWPE MPs collected after reaction in a control CN/H₂O₂ system at 140 °C. Reaction conditions: [UHMWPE MPs] = 1 g L⁻¹, [CN] = 1 g L⁻¹, hydrothermal temperature = 140 °C and neutral pH.

Supplementary Fig. 27 | Photos of UHMWPE MPs after reaction in a pure H₂O system at 140 °C. Reaction conditions: [UHMWPE MPs] = 1 g L⁻¹, hydrothermal temperature = 140 °C and neutral pH.

Comment 3-3. The EPR spectrum of superoxide radicals should be provided. Other free radicals may be generated during the reaction process.

Response: Thank you for the valuable suggestion. Identifying the possible radicals and non-radicals generated during the reaction process is important to understand the reaction mechanism. DMPO spin-trapping EPR spectra for superoxide radical (O₂^{•-}) detection was performed in a methanol/water (10% methanol) mixed solution. No characteristic peak corresponding to O₂^{•-} radicals was detected. This indicated that H₂O₂ was activated by FeSA-hCN for the selective production of •OH.

Changes: *The following content was added on page 10. (Second paragraph)*

“To explore the ROS involved in FeSA-hCN/H₂O₂ system, electron paramagnetic resonance (EPR) and chemical quenching experiments were conducted (Fig. 5a and b, **Supplementary Figs. 40** and 41). The EPR analysis showed that FeSA-hCN/H₂O₂/DMPO system produced a 4-fold characteristic peak with an intensity ratio of 1:2:2:1, corresponding to •OH (Fig. 5a). **No O₂^{•-} was detected (Supplementary Fig.**

40). The quenching experiments confirmed the significant contribution of $\bullet\text{OH}$ on UHMWPE degradation (Fig. 5b and Supplementary Fig. 41).”

Supplementary Materials, page 42, Supplementary Fig. 40 was added.

Supplementary Fig. 40 | DMPO spin-trapping EPR spectra of superoxide radicals (O₂^{•-}) in a methanol/water (10% methanol) mixed solution.

Comment 3-4. The chemical formula of the reaction intermediate should be supplemented. In addition, the Gibbs free energy of Fe-C₃N₄ in steps 4 and 5 is higher than that of C₃N₄, which leads to a decrease in activity.

Response: Thank you for the excellent advice. The chemical formulas of reaction intermediates were supplemented. Although the free energy changes of Fe-C₃N₄ in steps 4 and 5 are higher than those of C₃N₄, the rate-determining step (RDS) of C₃N₄ requires a larger energy barrier (1.70 eV) for the generation of $\bullet\text{OH} + \bullet\text{OH}$, which permits the subsequent reaction pathways. For Fe-C₃N₄, the RDS is the formation of $\bullet\text{OH} + \bullet\text{OH}$, with a relatively low free energy barrier of 1.62 eV. In DFT calculations, we also took into account the adsorption energy of H₂O₂, the change of O–O bond length in H₂O₂, and the electronic structures and charge transfer, which worked together for efficient H₂O₂ activation by Fe-C₃N₄ from a theoretical perspective.

Changes: *The chemical formulas of the reaction intermediates were provided in Supplementary Figs. 48 and 49, Supplementary Materials, pages 50, 51.*

Supplementary Fig. 48 | Chemical structure of identified products from Ret time of 3.6 to 15.3 min (Supplementary Table 7).

Supplementary Fig. 49 | Chemical structure of identified products from Ret time of 15.3 to 23.8 min (Supplementary Table 7).

On page 12, corresponding changes were made. (Last paragraph)

“The reaction intermediates/products from UHMWPE degradation are identified by gas chromatography-mass spectrometry (GC-MS, Supplementary Figs. 48 and 49, Supplementary Tables 7 and 8).”

The following discussion was added on page 12 to theoretically highlight the superiority of Fe-C₃N₄ over C₃N₄ in H₂O₂ activation. (Second paragraph)

“The fast transfer rate between Fe single sites and H₂O₂ can impel the activation of H₂O₂⁴². In contrast, no noticeable change was observed on C₃N₄. From a theoretical perspective, the integration of obvious stretched O–O bond of H₂O₂, strong H₂O₂ adsorption, low energy barrier of RDS, and favorable electronic structure of Fe-C₃N₄ is responsible for the superior activity of FeSA-hCN in H₂O₂ activation to hCN or CN.”

Comment 3-5. During the reaction process, gas products will be generated, and the author should detect them.

Response: Thank you very much for the professional suggestion. We supplied the detection of gas products at 140 °C hydrothermal condition, by using an externally heated autoclave reactor equipped with gas sampling ports. Compared to UHMWPE/H₂O control systems, trace CO₂ and H₂ gases were detected in the UHMWPE/H₂O₂/FeSA-hCN system. Signals of acetylene and methane were also detected, but the intensities were lower than the analytical limits. The amount of produced gases is far lower than that of the organic chemicals produced in the solution detected by TOC. This further verified that UHMWPE MPs were mostly converted into organic carbon species.

Changes: *The following changes were made on page 8. (Second paragraph)*

“We also tried to detect gas products. Only trace CO₂ and H₂ were detected after the reaction, and their amounts were far lower than the organic chemicals in the solution (Supplementary Fig. 30). Signals of acetylene (C₂H₂) and methane (CH₄) were also found; however, their intensities were below the analytical limit (Supplementary Fig. 31). TOC and gas product analysis further verified that the UHMWPE MPs were mostly converted into organic chemicals.”

Page 19 (Last paragraph)

“Detection of gases produced during hydrothermal reaction

An externally heated autoclave reactor (Aulight CEL-MPR microreactor, Beijing) equipped with gas sampling ports was used. UHMWPE MPs (1 g L⁻¹) and 1 g L⁻¹ FeSA-hCN catalysts were added into the reactor with 50 mL ultrapure water. After that, 100 mM H₂O₂ was added. Then, the autoclave reactor was carefully sealed, and the operation temperature was set at 140 °C for 18 h. After the reaction, the reactor was allowed to naturally cool down to room temperature. A gas sampling needle is then employed to extract a gas sample from the sampling port and inject it into the gas chromatography (Agilent 8890) for analysis.”

Supplementary Materials, pages 32-33, Supplementary Figs. 30 and 31 were added.

Supplementary Fig. 30 | Detection of gases produced after the hydrothermal UHMWPE MPs degradation using the 8890 GC-TCD detector. Reaction conditions: reaction time = 18 h, [UHMWPE MPs] = 1 g L⁻¹, [catalysts] = 1 g L⁻¹ if used, [H₂O₂] = 100 mM if used, hydrothermal temperature = 140 or 160 °C and neutral pH.

After 18 h hydrothermal degradation of UHMWPE, the amount of CO₂ in the system was determined as 29.6 mg L⁻¹, and H₂ was detected as 0.4 mg L⁻¹.

Supplementary Fig. 31 | Detection of gases produced during hydrothermal UHMWPE MPs degradation using the 8890 GC-FID detector. Reaction conditions: reaction time = 18 h, [UHMWPE MPs] = 1 g L⁻¹ if present, [catalysts] = 1 g L⁻¹ if used, [H₂O₂] = 100 mM if used, hydrothermal temperature = 140 or 160 °C and neutral pH.

Only small signals of acetylene (C_2H_2) and methane (CH_4) were detected, which are below the analytical limit.

REVIEWERS' COMMENTS

Reviewer #1 (Remarks to the Author):

The manuscript has been revised as required, and I recommend accepting the paper.

Reviewer #2 (Remarks to the Author):

The issues raised by me were addressed by authors. I have no other more questions. The revised manuscript can be accepted for the publication.

Reviewer #3 (Remarks to the Author):

The revised manuscript has been updated according to the comments, and the present version can be accepted now.